# Generalized Precision Matrix for Scalable Estimation of Nonparametric Markov Networks

**Yujia Zheng**[1,2], **Ignavier Ng**[1], **Yewen Fan**[1], **Kun Zhang**[1,2]
[1] Carnegie Mellon University
[2] Mohamed bin Zayed University of Artificial Intelligence
`{yujiazh, ignavierng, yewenf, kunz1}@cmu.edu`

## Abstract

A Markov network characterizes the conditional independence structure, or Markov property, among a set of random variables. Existing work focuses on specific families of distributions (e.g., exponential families) and/or certain structures of graphs, and most of them can only handle variables of a single data type (continuous or discrete). In this work, we characterize the conditional independence structure in *general distributions for all data types* (i.e., continuous, discrete, and mixed-type) with a Generalized Precision Matrix (GPM). Besides, we also allow *general functional relations among variables*, thus giving rise to a Markov network structure learning algorithm in one of the most general settings. To deal with the computational challenge of the problem, especially for large graphs, we unify all cases under the same umbrella of a regularized score matching framework. We validate the theoretical results and demonstrate the scalability empirically in various settings.

## 1 Introduction

*Markov networks* (also known as *Markov random fields*) represent conditional dependencies among random variables. They provide clear semantics in a graphical manner to cope with uncertainty in probability theory, with a wide application in fields including physics (Cimini et al., 2019), chemistry (Dodani et al., 2016), biology (Jaimovich et al., 2006), and sociology (Carrington et al., 2005). The undirected nature of edges also allows cyclic, overlapping, or hierarchical interactions (Shen et al., 2009). To estimate the Markov network from observational data, existing work focuses on certain parametric families of distributions, a majority of which study the Gaussian case. By assuming that the variables are from a multivariate Gaussian distribution, the dependencies can be well represented by the support of the precision, or inverse covariance, matrix according to Hammersley-Clifford theorem (Besag, 1974; Grimmett, 1973). Together with various statistical estimators (e.g., the *graphical lasso* (Friedman et al., 2008) and *neighborhood selection* (Meinshausen & Bühlmann, 2006)), this connection between the precision matrix and graphical structure has been well exploited in the Gaussian case in the past decades (Yuan, 2010; Ravikumar et al., 2011). However, methods for Gaussian graphical models fail to correctly capture dependencies among variables deviating from Gaussian or including nonlinearity (Raskutti et al., 2008; Ravikumar et al., 2011).

While non-Gaussianity is more common in real-world data generating process, few results are applicable to Markov network structure learning on non-Gaussian data. In the *discrete* setting, Ravikumar et al. (2010) showed that a binary Ising model can be recovered by neighborhood selection using $\ell_1$ penalized logistic regression. Loh & Wainwright (2013) encoded extra structural relations in the proposed generalized covariance matrix to model the dependencies for Markov networks with certain structures (e.g., tree structures or graphs with only singleton separator sets) among variables from exponential families. Several approaches allowed estimation for non-Gaussian continuous variables while most of them assumed parametric assumptions such as the exponential families (Yang et al., 2015; Lin et al., 2016; Suggala et al., 2017) or Gaussian copulas (Liu et al., 2009; 2012; Harris & Drton, 2013). These methods illustrate the possibility of reliable Markov network estimations in several non-Gaussian cases, but still, the models are restricted to specific parametric families of distributions and/or structures of conditional independences.

Concerned with describing Markov properties of non-Gaussian data with general *continuous* distributions, Morrison et al. (2017) used the second-order derivatives to encode the conditional independence structure. Specifically, their approach is based on a theorem that the zero pattern in the Hessian matrix of the log-density determines the conditional independencies between non-Gaussian continuous variables (Spantini et al., 2018). A method based on transport map, i.e., Sparsity Identification in Non-Gaussian distributions (SING) (Baptista et al., 2021), is then designed to estimate the data density from samples, and the structure is derived from the estimated density. This approach achieves consistent Markov network structure recovery in a general non-Gaussian continuous setting. However, methods relying on the Hessian matrix cannot cope with discrete or mixed-type data. In addition, density estimation, especially for non-Gaussian data, can be computationally challenging for large graphs, limiting the scalability of this approach. Kernel-based Conditional Independence test (KCI) (Zhang et al., 2012) and Generalized Score (GS) (Huang et al., 2018) can handle the mixed-type case for structure learning, but as kernel-based methods, they are computationally challenging since the complexity scales cubically in the number of samples.

To deal with these remaining obstacles, we explore a Generalized Precision Matrix (GPM) for nonparametric Markov networks learning. Based on the necessary and sufficient conditions for the conditional independence among structures in continuous, discrete, and mixed-type cases, GPM characterizes the Markov network structures with arbitrary data types. Moreover, our work does not constrain the distribution to be of specific families, such as exponential families, or has been normalized. Besides, it is also noteworthy that there are no specific assumptions on the functional relations among variables. To the best of our knowledge, the proposed GPM illustrates the feasibility of Markov network structure learning in one of the most general nonparametric settings.

Furthermore, we put all these cases under the same umbrella of the estimation framework based on regularized score matching, as an extension of the score matching framework (Hyvärinen & Dayan, 2005). Different from the previous approach (SING) that applies a transport map to estimate the data density for general continuous distributions, our framework allows us to only estimate the model score function parameterized by a deep model, from which the characterization matrix of the Markov network structure can be directly calculated. To facilitate the estimation process, we also exploit suitable penalties on the characterization matrix to encourage constantly sparse entries. Besides, we adopt recent advancements on score matching (Song et al., 2020) to further scale up the process. Our method therefore narrows the gap between reliable structure learning and scalable deep learning techniques. We validate the theoretical results experimentally, and the scalability has been illustrated.

## 2 GENERALIZED PRECISION MATRIX

Suppose that we observe a collection of random variables $\mathbf{X} = (X_1, \ldots, X_d)$. Our goal is to discover the underlying Markov network structure. Specifically, it is an undirected graph $\mathcal{G}$ comprising a set of vertices $\mathbf{V} = \{1, \ldots, d\}$ and edges $\mathbf{E}$. The edges $\mathbf{E}$ encode the conditional independence relations or the global Markov property: for any disjoint subsets $\mathbf{A}$, $\mathbf{B}$, and $\mathbf{C}$ in the vertices set $\mathbf{V}$ such that $\mathbf{C}$ separates $\mathbf{A}$ and $\mathbf{B}$, $\mathbf{X_A}$ and $\mathbf{X_B}$ are conditionally independent given $\mathbf{X_C}$, i.e., $\mathbf{X_A} \perp\!\!\!\perp \mathbf{X_B} \mid \mathbf{X_C}$.[1] Throughout this paper, we use an uppercase letter to denote a random variable and a lowercase letter with subscripts to denote the value of a random variable (e.g., $X_i = x_i$ for the value of $X_i$). For a discrete variable, say $X_i$, we denote its support by $\{x_{i1}, \ldots, x_{iM_i}\}$, where $M_i$ is its cardinality.

As an alternative characterization of the conditional independence relations encoded by the graph, the pairwise Markov property requires that every pair of non-adjacent variables in the graph is conditionally independent given the remaining variables. That is, for any $i \neq j$, an edge between $X_i$ and $X_j$ is absent if and only if $X_i$ and $X_j$ are conditionally independent given the remaining variables, i.e., $X_i \perp\!\!\!\perp X_j \mid \mathbf{X}_{\mathbf{V} \setminus \{i,j\}}$. The conditioning set consisting of all remaining variables is essential. According to Lauritzen (1996), the pairwise Markov property is equivalent to the global one when the density is strictly positive. In order to estimate nonparametric Markov networks in this setting, we explore generalized characterizations of conditional independence in all types of data (i.e., continuous, discrete, and mixed-type) without distributional constraints. We start from learning conditional independence structures in continuous data with a procedure inspired by Spantini et al. (2018), and then propose new characterizations for discrete and mixed-type data.

---

[1] For any set $\mathbf{S} \subset \mathbf{V}$, we write $\mathbf{X_S} = \{X_i : i \in \mathbf{S}\}$.

Ideally, we aim to construct a Generalized Precision Matrix $\Omega$ that satisfies the following desiderata:

a. For any $i \neq j$, if $\Omega_{i,j} = 0$, then $X_i \perp\!\!\!\perp X_j \mid X_{\mathbf{V}\setminus\{i,j\}}$;

b. The probability measure is *not restricted* to be from specific families but only needs to be strictly positive;

c. The undirected graph $\mathcal{G}$ is *not restricted* to be of certain structures;

d. For continuous variables, the density has continuous derivatives up to second order w.r.t. the Lebesgue measure;

e. For discrete variables, the cardinality is not restricted;

f. To enable practical estimation procedure, $\Omega$ is differentiable w.r.t. $\mathbf{X}$.

Property (a) is the characterization of the pairwise Markov property. Properties (b) and (c) differentiate our work from most previous works that assumes Gaussianity or/and certain structures of the conditional independence. Properties (d) and (e) further raise the difficulty of our task, because, in addition to not being restricted to a specific family of distributions, our characterization $\Omega$ has to be available for all data types (i.e., continuous, discrete, and mixed-type) with mild assumptions. For discrete variables, Property (e) removes the limitation of cardinality, thus differentiating our work from those focusing on the binary Ising model. Property (f) allows us to incorporate an $\ell_1$ regularization term in the estimation procedure and make use of gradient-based optimization.

## 2.1 CHARACTERIZATION FOR CONTINUOUS DATA

We aim to find the necessary and sufficient conditions for $X_i \perp\!\!\!\perp X_j \mid \mathbf{X}_{\mathbf{V}\setminus\{i,j\}}$. By definition, if $X_i$ is conditionally independent of $X_j$ given all remaining variable $\mathbf{X}_{\mathbf{V}\setminus\{i,j\}}$, we can factor the probability density function (PDF) $p_{\mathbf{X}}$ as follows

$$p_{\mathbf{X}}(\mathbf{x}) = p(x_i \mid \mathbf{x}_{\mathbf{V}\setminus\{i,j\}})p(x_j \mid \mathbf{x}_{\mathbf{V}\setminus\{i,j\}})p(\mathbf{x}_{\mathbf{V}\setminus\{i,j\}}). \tag{1}$$

Together with the assumption that $p_{\mathbf{X}}$ has continuous derivatives up to second order w.r.t. the Lebesgue measure, we have

$$\frac{\partial^2 \log p_{\mathbf{X}}}{\partial x_i \partial x_j} = 0. \tag{2}$$

Conversely, the solution of Eq. (2) is given by $\log p_{\mathbf{X}}(\mathbf{x}) = g(x_{1:i-1}, x_{i+1:d}) + h(x_{1:j-1}, x_{j+1:d})$ for some functions $g, h : \mathbb{R}^{d-1} \to \mathbb{R}$. It thus follows $X_i \perp\!\!\!\perp X_j \mid \mathbf{X}_{\mathbf{V}\setminus\{i,j\}}$. This connection between pairwise conditional independence and cross derivatives of the log density has been observed in Spantini et al. (2018). Methods based on this connection have also been proposed recently (Morrison et al., 2017; Baptista et al., 2021). Following Baptista et al. (2021), one can characterize the conditional independence between $X_i$ and $X_j$ in the continuous distribution as

$$\Omega_{ij}^{[c]} := \left( \mathbb{E}_{p_{\mathbf{X}}} \left[ f_{i,j}^{[c]}(\mathbf{x})^2 \right] \right)^{\frac{1}{2}}, \tag{3}$$

where $f_{i,j}^{[c]}(\cdot)$ denotes the LHS of Eq. (2) and $[c]$ denotes continuous data as a type label. In practice, $p_{\mathbf{X}}$ is the empirical PDF. The group structure of it could help achieve simultaneous sparse approximation (Yuan & Lin, 2006; Huang & Zhang, 2010) when being applied as an $\ell_1$ regularizer in the estimation, which we will describe in Sec. 3. We also apply the same group structures for both the discrete and mixed-type cases, but we will skip the reintroduction for brevity. The characterization of the Markov property is as follows.

**Corollary 1.** *Assume*

*i.* $\mathbf{X} = (X_1, \ldots, X_d)$ *is a set of continuous variable.*

*ii. The PDFs of $\mathbf{X}$ are strictly positive and smooth.*

*iii. The characterization matrix $\Omega^{[c]}$ is defined according to Eq. (3).*

*Then for any $i \neq j$, $\Omega_{i,j}^{[c]} = 0$ implies $X_i \perp\!\!\!\perp X_j \mid \mathbf{X}_{\mathbf{V}\setminus\{i,j\}}$.*

The proof is shown in Appx. A.1. It is worth noting that Cor. 1 also covers the Gaussian case, where the cross-derivatives of the log-density correspond to entries in the precision or inverse covariance matrix (Drton et al., 2008), thus generalizing previous work assuming Gaussianity. Hence, the support of $\Omega$ characterizes conditional independence among continuous variables for general distributions.

## 2.2 CHARACTERIZATION FOR DISCRETE DATA

Since most of the previous work focuses on the Gaussian setting, and works for non-Gaussian distribution are mostly restricted to the exponential family, the characterization for continuous data discussed in Sec. 2.1 has broadened the scope of reliable Markov network learning. However, the characterization is not applicable to discrete data as the gradient does not exist. In this section, we provide such a characterization of Markov network structure in the discrete case. Similar to the continuous case, a key ingredient of the proposed characterization is the necessary and sufficient conditions of conditional independence for discrete data, which we establish in the following theorem.

**Theorem 1.** *Denote* $\mathbf{V}$ *as a set of discrete variables and* $X_i, X_j \in \mathbf{V}$*. For brevity, denote* $\mathbf{V} \backslash \{X_i, X_j\}$ *as* $\mathbf{Z}$*. Let* $\{x_{i1}, \ldots, x_{iM_i}\}$ *and* $\{x_{j1}, \ldots, x_{jM_j}\}$ *be the support of variables* $X_i$ *and* $X_j$*. Denote* $z$ *as any value(s) of* $\mathbf{Z}$*. Then,* $X_i \perp\!\!\!\perp X_j \mid \mathbf{Z}$ *if and only if, for all* $k \in [M_i]$ *and* $l \in [M_j]$ *with* $k \neq 1$ *and* $l \neq 1$*, we have*

$$(\log m(x_{i1}, x_{j1}, z) - \log m(x_{ik}, x_{j1}, z)) - (\log m(x_{i1}, x_{jl}, z) - \log m(x_{ik}, x_{jl}, z)) = 0. \quad (4)$$

*Proof sketch.* For the sufficient condition, we want to show that the general solution to Eq. 4 has no term that takes the values of both $X_i$ and $X_j$. We first iterate all possible differences w.r.t. $X_j$ to get the discrete score function of $X_i$, which does not take the value of $X_j$ as the argument. Then we obtain the desired solution by summation over all possible differences w.r.t. $X_i$. For the necessary condition, we decompose the PMF according to the conditional independence to obtain Eq. 4.

The full proof is provided in Appx. A.2. Note that we denote $m(x_{ik}, x_{jl}, z)$ as the joint probability mass function (PMF) of $\{X_i, X_j, \mathbf{Z}\}$, simplified from $m_{X_i, X_j, \mathbf{Z}}(x_{ik}, x_{jl}, z)$. Based on Thm. 1, we propose the characterization matrix of conditional independence for discrete data $\Omega_{i,j}^{[d]}$ as follows:

$$\Omega_{i,j}^{[d]} := \mathbb{E}_{m_{\mathbf{X}}} \left[ \sum_{k,l} f^{[d]}(x_{i1}, x_{ik}, x_{j1}, x_{jl}, z)^2 \right], \quad (5)$$

where $f^{[d]}(x_{i1}, x_{ik}, x_{j1}, x_{jl}, z)$ denotes the LHS of Eq. (4) and $[d]$ is a type label denoting discrete data. The support of the matrix above satisfies the pairwise Markov property and characterizes the Markov network structure, formally stated below with its proof in Appx. A.3.

**Corollary 2.** *Assume*

    *i.* $\mathbf{X} = (X_1, \ldots, X_d)$ *is a set of discrete variable.*

    *ii. The PMFs of* $\mathbf{X}$ *are strictly positive.*

    *iii. The characterization matrix* $\Omega^{[d]}$ *is defined according to Eq.* (5).

*Then for any* $i \neq j$*,* $\Omega_{i,j}^{[d]} = 0$ *implies* $X_i \perp\!\!\!\perp X_j \mid X_{\mathbf{V} \backslash \{i,j\}}$*.*

Therefore, we have a characterization matrix $\Omega^{[d]}$ to represent the conditional independence structure for discrete data. It is worth noting that, unlike the generalized covariance matrix in Loh & Wainwright (2012) that only applies to certain structures among variables from exponential families, the proposed characterization matrix $\Omega^{[d]}$ encodes the Markov properties for general discrete distributions without any structural constraints. Also, compared with Ravikumar et al. (2010), Thm. 1 can be applied to general graphical models apart from binary Ising models and does not rely on the structural condition. It also does not limit the cardinalities of discrete variables. Hence, Theorem 1 sheds light on characterizing arbitrary conditional independence structures for general discrete distributions.

## 2.3 CHARACTERIZATION FOR MIXED-TYPE DATA

In the previous sections, we have presented characterizations of conditional independence structures for both general continuous and discrete distribution. However, it is common for real-world datasets to have a mixture of continuous and discrete variables. Unfortunately, most works focus on either continuous or discrete data, and previous results for mixed-type data are mostly based on conditional Gaussian distribution (Lauritzen et al., 1989; Edwards, 1990; Lauritzen, 1996; Fellinghauer et al., 2013; Lee & Hastie, 2015; Cheng et al., 2017). Similar to the continuous and discrete settings,

in this section, we introduce a novel characterization of the pairwise Markov property for general distributions with mixed data-types. We first provide necessary and sufficient conditions of conditional independence for mixed-type data in the following theorem, with full proof given in Appx. A.4.

**Theorem 2.** *Denote $\mathbf{V}$ as a set of mixed-type variables and $X_i, X_j \in \mathbf{V}$, where $X_i$ is discrete and $X_j$ is continuous. Let $\{x_{i1}, \dots, x_{iM_i}\}$ be the support of variables $X_i$. For brevity, denote $\mathbf{V} \backslash \{X_i, X_j\}$ as $\mathbf{Z}$. Denote $z$ as any value(s) of $\mathbf{Z}$ and $x_j$ as any value of the continuous variable $X_j$. Then, $X_i \perp\!\!\!\perp X_j \mid \mathbf{Z}$ if and only if, for all $k \in [M_i]$ with $k \neq 1$, we have*

$$\frac{\partial \log \left( p_{X_j, \mathbf{Z} \mid X_i}(x_j, z \mid x_{i1}) m_{X_i}(x_{i1}) \right)}{\partial x_j} - \frac{\partial \log \left( p_{X_j, \mathbf{Z} \mid X_i}(x_j, z \mid x_{ik}) m_{X_i}(x_{ik}) \right)}{\partial x_j} = 0. \tag{6}$$

*Proof sketch.* Similar to the proof sketch of Thm. 1, we consider $X_i$ and $X_j$ separately to construct the desired general solution of Eq. 6 for the sufficient condition. For the necessary condition, we decompose the density function according to the conditional independence to obtain Eq. 6.

Based on Thm. 2, we propose to characterize the conditional independence between $X_i$ and $X_j$ given all remaining variables with the GPM $\Omega^{[m]}$, of which the element is defined as

$$\Omega_{i,j}^{[m]} := \begin{cases} \mathbb{E}_{\pi_\mathbf{x}} \left[ f_{i,j}^{[c]}(\mathbf{x})^2 \right] & \text{if } X_i \in \mathbf{X}_c, X_j \in \mathbf{X}_c \\ \mathbb{E}_{\pi_\mathbf{x}} \left[ \sum_{k,l} f^{[d]}(x_{i1}, x_{ik}, x_{j1}, x_{jl}, z)^2 \right] & \text{if } X_i \in \mathbf{X}_d, X_j \in \mathbf{X}_d \\ \mathbb{E}_{\pi_\mathbf{x}} \left[ \sum_k f^{[m]}(x_i, x_{j1}, x_{jk}, z)^2 \right] & \text{if } X_i \in \mathbf{X}_c, X_j \in \mathbf{X}_d \\ \mathbb{E}_{\pi_\mathbf{x}} \left[ \sum_k f^{[m]}(x_j, x_{i1}, x_{ik}, z)^2 \right] & \text{if } X_i \in \mathbf{X}_d, X_j \in \mathbf{X}_c, \end{cases} \tag{7}$$

where $f^{[m]}$ denotes LHS of Eq. (6) and $\pi_\mathbf{X}$ is the probability function. The type label $[m]$ denotes mixed-type data. $\mathbf{X}_c$ and $\mathbf{X}_d$ are sets of continuous and discrete variables, respectively. Its characterization of Markov property is as follows

**Corollary 3.** *Assume*

- *i. $\mathbf{X} = (X_1, \dots, X_d)$ is a set of variables containing both continuous and discrete variables.*

- *ii. For continuous variables, the PDFs are strictly positive and smooth.*

- *iii. For discrete variables, the PMFs are strictly positive.*

- *iv. The characterization matrix $\Omega^{[m]}$ is defined according to Eq. (7).*

*Then for any $i \neq j$, $\Omega_{i,j}^{[m]} = 0$ implies $X_i \perp\!\!\!\perp X_j \mid \mathbf{X}_{\mathbf{V} \backslash \{i,j\}}$.*

The proof is given in Appx. A.5. The GPM $\Omega^{[m]}$ encodes the pairwise Markov property for mixed-type data. More general than previous works, it does not require specific families of distributions, structures of the underlying graph, or cardinality of discrete variables.

## 3    SCALABLE ESTIMATION WITH REGULARIZED SCORE MATCHING

In Sec. 2, we provide characterizations of conditional independencies for general distribution in continuous, discrete, and mixed-type settings. Based on the introduced necessary and sufficient conditions, these characterizations generalize previous work and establish one of the foundations for nonparametric estimation of Markov network structures with minimal assumptions.

In addition to general characterizations of the Markov property with theoretical guarantees (i.e., GPM), a scalable estimation framework is necessary for reliable and practical structure learning. Ideally, we would like to exploit the advancements on scalable deep learning models. Hence, we introduce a regularized score matching-based framework for all considered settings (i.e., general distributions of continuous, discrete, and mixed-type variables).

## 3.1 ESTIMATION FOR CONTINUOUS DATA

We start with the continuous setting. Denote $p(\mathbf{x}; \theta)$ as a parameterized density model with a parameter vector $\theta$. The goal is to estimate parameter $\theta$ from the observation $\mathbf{x}$. We aim to optimize the following objective function, which is based on Fisher divergence:

$$O_c(\theta) = \frac{1}{2} \int_{\mathbf{x} \in \mathbb{R}^d} p(\mathbf{x}) \|\nabla_{\mathbf{x}} \log p(\mathbf{x}; \theta) - \nabla_{\mathbf{x}} \log p(\mathbf{x})\|^2 d\mathbf{x} + \rho_\lambda(\Omega^{[c]}), \tag{8}$$

where $\rho_\lambda(\cdot)$ denotes a sparsity penalty function and $\lambda$ is the penalty parameter with domain $[0, 1]$. $\Omega^{[c]}$ is defined in Eq. 3 as our characterization of the conditional independence structure for continuous data. If we assume the model is not degenerate, where different values of $\theta$ correspond to different PDFs, the asymptotic consistency of the optimization has been shown in Thm. 2 by Hyvärinen & Dayan (2005). We impose a sparsity penalty to encounter for finite-sampling errors in practice.

Also with a strategy in Hyvärinen & Dayan (2005); Pham & Garat (1997), one can remove the data log-density $\log p_{\mathbf{X}}$ from Eq. (8) by optimizing the following equation, which is equivalent to Eq. (8):

$$O_c(\theta) = \int_{\mathbf{x} \in \mathbb{R}^d} p(\mathbf{x}) \sum_{i=1}^{d} \left[ \frac{1}{2} \|\nabla_{x_i} \log p(\mathbf{x}; \theta)\|^2 + H_{x_i}(\log p(\mathbf{x}; \theta)) \right] d\mathbf{x} + \rho_\lambda(\Omega^{[c]}), \tag{9}$$

where $H$ denotes the Hessian. The proof is directly based on Hyvärinen & Dayan (2005) and we include it (Lemma 1) in Appx. A.6.1 for completeness. It is worth noting that previous work on Markov network structure learning with general continuous distribution (SING (Morrison et al., 2017; Baptista et al., 2021)) applies a transport map to estimate data density from samples, which can be computationally challenging for non-Gaussian data with a large number of variables. Thus, it may not be scalable as suggested by Fig. 1 and Table 1. To avoid this, the proposed regularized score matching allows us to optimize the objective function by only estimating the model score function. Moreover, the estimated model score function directly leads to the characterization matrix $\Omega^{[c]}$ by taking further derivatives, thus efficiently giving rise to the estimated Markov network structure. After training, the expectation in Equation 3 is computed over the parameterized model $p(\mathbf{x}; \theta)$.

## 3.2 ESTIMATION FOR DISCRETE DATA

For the estimation in the discrete case, one cannot directly apply the method introduced for the continuous case since the gradient, on which the continuous score function is based, is not defined for discrete data. An intuitive solution is to replace the gradient with a general linear operator $\mathcal{L}$ (Lyu, 2012). Of course, one also needs to replace integration with summation and PDF with PMF. For instance, Eq. (8) can be reformulated as follows

$$O_d(\theta) = \frac{1}{2} \sum_{\mathbf{x}} m_{\mathbf{X}}(\mathbf{x}) \left\| \frac{\mathcal{L}(m(\mathbf{x}; \theta))}{m(\mathbf{x}; \theta)} - \frac{\mathcal{L}(m_{\mathbf{X}}(\mathbf{x}))}{m_{\mathbf{X}}(\mathbf{x})} \right\|^2 + \rho_\lambda(\Omega^{[d]}), \tag{10}$$

where $m$ denotes PMF. In this formulation, $\mathcal{L}(\cdot)$ is a generalized version of the score function for discrete data. As shown in Lyu (2012), $O_d(\theta)$ keeps the computational advantages of score matching for continuous data, i.e., the normalizing partition is canceled out and the formulation can be transformed to an expectation of functions of the unnormalized model. In order to guarantee the consistency of score matching based on Eq. (10), the linear operator $\mathcal{L}(\cdot)$ needs to be *complete* according to the following definition.

**Definition 1 (Completeness (Lyu, 2012)).** *A linear operator $\mathcal{L}(\cdot)$ is complete if $\frac{\mathcal{L}(p(\mathbf{x}))}{p(\mathbf{x})} = \frac{\mathcal{L}(q(\mathbf{x}))}{q(\mathbf{x})}$ implies $p(\mathbf{x}) = q(\mathbf{x})$ almost everywhere, where $p(\mathbf{x})$ and $q(\mathbf{x})$ are two PMFs.*

According to Defn. 1, Lyu (2012) used the marginalization operator $\mathcal{M}(\cdot) : \mathcal{F}^1 \mapsto \mathcal{F}^d$ as a choice for $\mathcal{L}(\cdot)$, which is defined as

$$\mathcal{M}(f(\mathbf{x})) = \begin{pmatrix} \vdots \\ \mathcal{M}_i(f(\mathbf{x})) \\ \vdots \end{pmatrix} = \begin{pmatrix} \vdots \\ \sum_{\mathbf{x}} f(\mathbf{x}) \\ \vdots \end{pmatrix}, \tag{11}$$

where $f \in \mathcal{F}^1$. We can observe that $\mathcal{M}_i(f(\mathbf{x}))$ is the marginal density of $\mathbf{x}^{\backslash i}$, where $\mathbf{x}^{\backslash i}$ denotes the vector $\mathbf{x}$ after dropping the $i$-th element (i.e., marginalization). The completeness of $\mathcal{M}(\cdot)$ has been shown in Brook (1964), and included as Lemma 3 in Lyu (2012). We have

$$O_d(\theta) = \frac{1}{2} \sum_{\mathbf{x}} m_{\mathbf{X}}(\mathbf{x}) \left\| \frac{\mathcal{M}(m(\mathbf{x};\theta))}{m(\mathbf{x};\theta)} - \frac{\mathcal{M}(m_{\mathbf{X}}(\mathbf{x}))}{m_{\mathbf{X}}(\mathbf{x})} \right\|^2 + \rho_\lambda(\Omega^{[d]}). \tag{12}$$

Thus, it is plausible for us to replace the gradient with $\mathcal{M}(\cdot)$ for discrete data. However, one key advantage of regularized score matching is that it does not have to explicitly estimate the data density (i.e., $p_X(\mathbf{x})$ in Theorem 1). As shown by Lyu (2012), we can also optimize Eq. (12) in a similar way, which is equivalent to optimizing the following equation

$$O_d(\theta) = \frac{1}{2} \sum_{\mathbf{x}} m_{\mathbf{X}}(\mathbf{x}) \sum_{i=1}^{d} \left[ \left( \frac{\mathcal{M}_i(m(\mathbf{x};\theta))}{m(\mathbf{x};\theta)} \right)^2 - 2\mathcal{M}_i \left( \frac{\mathcal{M}_i(m(\mathbf{x};\theta))}{m(\mathbf{x};\theta)} \right) \right] + \rho_\lambda(\Omega^{[d]}). \tag{13}$$

The simplification is directly from results in Lyu (2012), of which the corresponding lemma (Lemma 2) is formalized with its proof in Appx. A.6.2 for completeness. Based on Thm. 1 and Thm. 2, similar to the continuous case, we can estimate Markov network structures for general distributions in the discrete setting under the same umbrella of regularized score matching.

## 3.3 ESTIMATION FOR MIXED-TYPE DATA

For mixed-type data, we define the objective function as follows

$$O_m(\theta) = \mathbb{E}_{\pi_{\mathbf{X}}} \left[ \sum_i s_i(\mathbf{x};\theta) \right] + \rho_\lambda(\Omega^{[m]}), \tag{14}$$

where

$$s_i(\mathbf{x};\theta) := \begin{cases} \frac{1}{2}\|\nabla_{x_i} \log \pi(\mathbf{x};\theta)\|^2 + H_{x_i}(\log \pi(\mathbf{x};\theta)) & X_i \in \mathbf{X}_c \\ \frac{1}{2}\left( \frac{\mathcal{M}_i(m(\mathbf{x};\theta))}{m(\mathbf{x};\theta)} \right)^2 - \mathcal{M}_i\left( \frac{\mathcal{M}_i(m(\mathbf{x};\theta))}{m(\mathbf{x};\theta)} \right) & X_i \in \mathbf{X}_d, \end{cases} \tag{15}$$

Here, the density $\pi$ is strictly positive. Basically, $O_m(\theta)$ is a regularized version of the combination of the objective functions for the continuous and discrete cases. Because $\Omega^{[m]}$ also encodes the dependencies between continuous and discrete variables, we can estimate its support for mixed-type data without assuming group structures of data types. The following corollary guarantees the consistency, where we define $O'_m(\theta)$ as $O_m(\theta) - \rho_\lambda(\Omega^{[m]})$.

**Corollary 4.** *Assume*

   i. *The data density $\pi_{\mathbf{X}}(\cdot)$ is equal to $\pi(\cdot;\theta^*)$ for some $\theta^*$.*

   ii. *The data density $\pi_{\mathbf{X}}(\cdot)$ and model density $\pi(\cdot;\theta)$ are strictly positive. $\pi_{\mathbf{X}}(\cdot)$ and $\pi(\cdot;\theta)$ is differentiable and twice-differentiable, respectively, w.r.t. continuous variables. For some $\theta^*$, $\pi_{\mathbf{X}}(\cdot) = \pi(\cdot;\theta^*)$ and no other parameter value gives a density that is equal to $\pi(\cdot;\theta^*)$ almost everywhere.*

   iii. *The expectations $E_{\pi_{\mathbf{X}}}\left[\|\log \pi(\mathbf{x};\theta)\|^2\right]$ and $E_{\pi_{\mathbf{X}}}\left[\|\log \pi_{\mathbf{X}}(\mathbf{x})\|^2\right]$ are finite for any $\theta$, and $\pi_{\mathbf{X}}(\mathbf{x}) \log \pi(\mathbf{x};\theta)$ goes to zero for any $\theta$ when $\|\mathbf{x}\| \to \infty$.*

*Then $O'_m(\theta) = 0$ implies $\theta = \theta^*$.*

Cor. 4 follows from Lemma 1 (Hyvärinen & Dayan, 2005) and Lemma 2 (Lyu, 2012), which are included in Appx. A.6. Together with Thm. 2, one can estimate Markov network structures for mixed-type data in a general setting.

## 3.4 SPARSITY REGULARIZATION

By minimizing the objective function $O(\theta) \in \{O_c(\theta), O_d(\theta), O_m(\theta)\}$, our goal is to essentially perform a model selection task, i.e., to learn of the support of $\Omega \in \{\Omega^{[c]}, \Omega^{[d]}, \Omega^{[m]}\}$. Here, using

$\ell_0$ penalty may be computationally infeasible because it leads to a discrete optimization problem that is difficult to solve. Following previous works (Tibshirani, 1996), we adopt the $\ell_1$ regularizer $\rho_\lambda(\Omega) = \lambda\|\Omega\|_1$. In particular, the high-dimensional support recovery of $\ell_1$ regularizer has been extensively studied in the literature; for instance, see Wainwright (2009) for variable selection and Ravikumar et al. (2008) for Gaussian graphical model selection. Although $\ell_1$ regularizer induces sparsity, it may lead to bias in the resulting solution and thereby worsen the performance (Fan & Li, 2001; Breheny & Huang, 2011). This is because the $\ell_1$ norm increases linearly with the absolute value of nonzero entries, which is different from $\ell_0$ norm that is constant for nonzero entries. Therefore, we experiment with smoothly clipped absolute deviation (SCAD) penalty (Fan & Li, 2001), minimax concave penalty (MCP) (Zhang, 2010), and adaptive $\ell_1$ penalty (Zou, 2006) in this work, which helps remedy the bias issue of $\ell_1$ regularization. Specifically, SCAD and MCP penalties may be interpreted as a hybrid of $\ell_0$ and $\ell_1$ penalties, while adaptive $\ell_1$ penalty reweighs the penalty coefficient $\lambda$ by the initial estimate of $\Omega$ without regularization. Furthermore, the support recovery of $\ell_1$ penalty relies on the incoherence condition in various cases (Wainwright, 2009; Ravikumar et al., 2008; 2011), which may be a rather strong assumption in practice, whereas the SCAD and MCP penalties do not (Loh & Wainwright, 2017). Thus, we adopt the SCAD penalty according to experimental results (Fig. 7 in Appx. B). We integrate the SCAD penalties for all cases but only introduced here for brevity.

## 4 EXPERIMENTS

**Setup.** We conduct experiments on two sets of distributions: (1) *Butterfly distributions* (Morrison et al., 2017; Baptista et al., 2021) and (2) *distributions from random graphs*. For *Butterfly distribution* in the continuous setting, we have $r$ i.i.d. pairs of random variables $(P_i, Q_i)$ defined as $P_i \sim \mathcal{N}(0,1)$ and $Q_i = W_i P_i$ with $W_i \sim \mathcal{N}(0,1)$ and $W_i \perp\!\!\!\perp P_i$. We replace the Gaussian distribution with the Multinomial distribution for the discrete case and mix the two different types of pairs for the mixed-type case with uniformly sampled proportion. For *distributions from random graphs*, we first generate a random decomposable directed acyclic graph. Then, for the continuous case, the data are sampled from nonlinear structural equation models (SEMs) with exogenous noises from an exponential distribution. We employ a multilayer perceptron (MLP) with randomly generated weights as the nonlinear function. For the discrete case, variables are generated via randomly parameterized Multinomial distributions of the variable being simulated and the discrete parents (Andrews et al., 2018). For the mixed-type case, we simulate data with the process described in Andrews et al. (2018), of which the details are included in Appx. B. Finally, we moralize all random decomposable DAGs to obtain the ground-truth Markov network structures. We use the deep kernel exponential family (DKEF) during estimation and optimize the objective function by gradient descent with the Adam optimizer. All experiments are on 12 CPU cores with 24 GB RAM.

**Considered methods.** We consider the following representative methods for comparison: *(KCI)* We adopt Kernel-based Conditional Independence test (KCI) (Zhang et al., 2012) with the Incremental Association Markov Blanket (IAMB) algorithm (Tsamardinos et al., 2003) to learn the Markov network structure in our settings. *(GS)* We denote GS as Greedy Equivalence Search (GES) (Chickering, 2002) with Generalized Score (GS). Because this procedure estimates causal structures represented by completed partially DAGs (CPDAGs), we moralize the results to obtain the Markov network structures. *(SING)* Sparsity Identification in Non-Gaussian distributions (SING) (Morrison et al., 2017; Baptista et al., 2021) is an algorithm designed for the estimation of Markov networks in non-Gaussian continuous distributions. It applies a transport map to estimate the data density. *(GLASSO)* Graphical Lasso (GLASSO) is a classical sparse penalized estimator for the inverse covariance matrix. *(NPN)* GLASSO with the nonparanormal transformation (Liu et al., 2009).

**Results.** We first conduct comparisons in general distributions for all data types (i.e., discrete, continuous, and mixed-type) with different numbers of variables and a sample size of 1000. Among the considered

Table 1: Running time for 12 variables.

| Method | Ours | KCI | GS | SING | GLASSO | NPN |
|--------|------|-----|-----|------|--------|-----|
| Time (s) | 62.9 | 247 | 9536.5 | 4020.3 | 4.4 | 20.8 |

methods, both KCI and GS are available for the estimation of Markov network structures for general distributions with all data types. SING can only deal with continuous data and is therefore only applied in the continuous setting. We also include (semi)parametric methods (GLASSO and NPN) for baselines in the considered general settings. We use Hamming distance between the estimated graph and the ground truth graph as the metric. All results are from 5 trials with different random seeds. The missing results are either due to timeout (i.e., > 1 day) or OOM.

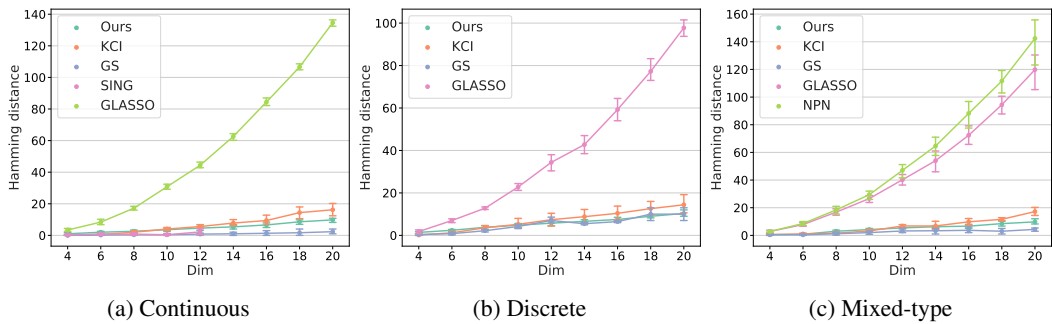

(a) Continuous                (b) Discrete                (c) Mixed-type

Figure 1: Hamming distances for Butterfly distributions.

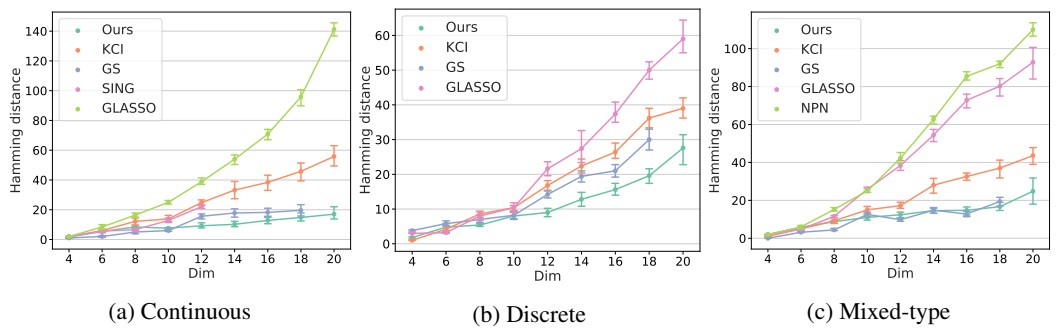

(a) Continuous                (b) Discrete                (c) Mixed-type

Figure 2: Hamming distances for distributions from random graphs.

For the Butterfly distributions (Fig. 1), one can observe that KCI, GS, and our method can almost recover the true structures with all data types. At the same time, in the more complex setting (i.e., distributions from random graphs, Fig. 2), it is clear that our method outperforms others in most datasets. This suggests that, compared to baselines, our method may have more obvious advantages in more complicated scenarios. Meanwhile, the running times of KCI, GS, and SING are significantly longer than that of our method (Table. 1). Besides, SING and GS cannot scale with more than 12 and 18 variables, respectively. GLASSO and NPN are remarkably fast but fail to accurately recover the structure in the general setting. NPN performs worse than GLASSO in structure recovery, which may be due to its misaligned hypothesis of the nonparanormal transformation in the general mixed-type setting.

We also conduct experiments on large graphs, with $\{250, 500, \ldots, 5000\}$ continuous variables from Butterfly distributions. Other settings are identical to those for smaller graphs. From Fig. 3, we observe that the running time is approximately linear w.r.t. the number of variables. Besides, all experiments are conducted on CPUs while our framework could be easily deployed on GPUs. This suggests the potential of taking advantage of recent advances in computation, especially for deep models, to even further improve the scalability.

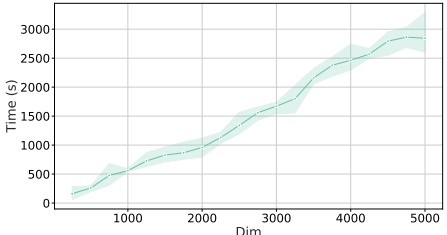

Figure 3: Running time for large graphs.

## 5 CONCLUSION

We provide a scalable estimation framework based on regularized score matching for nonparametric Markov network structures. We first introduce necessary and sufficient conditions of conditional independence among variables in general distributions for all data types (i.e., continuous, discrete, and mixed-type) without specific assumptions on functional relations among variables, thus giving rise to the corresponding characterizations of the structure, i.e., Generalized Precision Matrix. Then, we unify all these cases under the same umbrella of the estimation framework based on regularized score matching. Appropriate penalties on the characterization matrix are introduced to promote constantly sparse entries for stable estimation. We validate our theoretical claims experimentally in various settings. Future work includes exploring the connection between Markov networks and causal graphs.

## 6 ACKNOWLEDGMENT

We thank the anonymous reviewers for their constructive comments. This project was partially supported by the National Institutes of Health (NIH) under Contract R01HL159805, by the NSF-Convergence Accelerator Track-D award #2134901, by a grant from Apple Inc., a grant from KDDI Research Inc, and generous gifts from Salesforce Inc., Microsoft Research, and Amazon Research.

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

# Appendix

## Table of Contents

## A    PROOFS

### A.1    PROOF OF COROLLARY 1

**Corollary 1.** *Assume*

    *i.* $\mathbf{X} = (X_1, \ldots, X_d)$ *is a set of continuous variable.*

    *ii. The PDFs of* $\mathbf{X}$ *are strictly positive and smooth.*

    *iii. The characterization matrix* $\Omega^{[c]}$ *is defined according to Eq.* (3).

*Then for any* $i \neq j$, $\Omega^{[c]}_{i,j} = 0$ *implies* $X_i \perp\!\!\!\perp X_j \mid \mathbf{X}_{\mathbf{V} \setminus \{i,j\}}$.

*Proof.* According to Eq. (3), it is clear that when $\Omega^{[c]}_{i,j} = 0$, we have $\frac{\partial^2 \log p_\mathbf{X}}{\partial x_i \partial x_j} = 0$, which is a necessary and sufficient condition of $x_i \perp\!\!\!\perp x_j \mid x_{\mathbf{V} \setminus \{i,j\}}$ for strictly positive, smooth, and continuous distributions as shown in Sec. 2.1.

It is worth noting that it also applies in the Gaussian case, where $\mathbf{X} \sim \mathcal{N}(\mu, \Sigma)$ is a Gaussian vector with mean $\mu$ and non-singular covariance $\Sigma$. In this case, we have

$$P_\mathbf{X}(\mathbf{x}) \propto \exp\left( \frac{-(\mathbf{x} - \mu)^\top \Sigma^{-1} (\mathbf{x} - \mu)}{2} \right), \tag{16}$$

which implies

$$\frac{\partial^2 \log p_\mathbf{X}}{\partial x_i \partial x_j} = (\Sigma^{-1})^2_{i,j}, \tag{17}$$

where $(\Sigma^{-1})_{i,j}$ denotes the corresponding entry of the inverse covariance matrix. This well-known property of Gaussian distribution was also shown in Baptista et al. (2021); Drton et al. (2008). Because the inverse covariance matrix encodes the conditional independence structure when variables are from Gaussian distributions, $\Omega^{[c]}$ characterizes the Markov property for the Gaussian case. □

**Theorem 1.** *Denote* $\mathbf{V}$ *as a set of discrete variables and* $X_i, X_j \in \mathbf{V}$. *For brevity, denote* $\mathbf{V} \backslash \{X_i, X_j\}$ *as* $\mathbf{Z}$. *Let* $\{x_{i1}, \ldots, x_{iM_i}\}$ *and* $\{x_{j1}, \ldots, x_{jM_j}\}$ *be the support of variables* $X_i$ *and* $X_j$. *Denote* $z$ *as any value(s) of* $\mathbf{Z}$. *Then,* $X_i \perp\!\!\!\perp X_j \mid \mathbf{Z}$ *if and only if, for all* $k \in [M_i]$ *and* $l \in [M_j]$ *with* $k \neq 1$ *and* $l \neq 1$, *we have*

$$(\log m(x_{i1}, x_{j1}, z) - \log m(x_{ik}, x_{j1}, z)) - (\log m(x_{i1}, x_{jl}, z) - \log m(x_{ik}, x_{jl}, z)) = 0. \quad (4)$$

*Proof.* **Sufficient condition.**   Without loss of generality, let us consider three discrete variables, i.e., $\{X_i, X_j, Z\}$. Let $\{x_{i1}, \ldots, x_{iM_i}\}$ and $\{x_{j1}, \ldots, x_{jM_j}\}$ be the support of variables $X_i$ and $X_j$, respectively.

Consider the case that the finite difference of the discrete score function of $X_i$ w.r.t. $X_j$ equals zero.

When $X_i = x_{i1}$, $X_j = x_{j1}$, and differences are considered w.r.t. to $x_{ik'}$ and $x_{jl'}$, we have

$$\begin{aligned} &(\log m(x_{i1}, x_{j1}, z) - \log m(x_{ik'}, x_{j1}, z)) \\ &- (\log m(x_{i1}, x_{jl'}, z) - \log m(x_{ik'}, x_{jl'}, z)) = 0, \end{aligned} \quad (18)$$

where $m(x_{i1}, x_{j1}, z)$ is the joint PMF simplified from $m_{X_i, X_j, \mathbf{Z}}\{x_{i1}, x_{j1}, z\}$. By iterating all possible differences w.r.t. $X_j$, for all $l$ in $\{2, \ldots, M_j\}$, we have

$$\begin{aligned} &(\log m(x_{i1}, x_{j1}, z) - \log m(x_{ik'}, x_{j1}, z)) \\ &- (\log m(x_{i1}, x_{jl}, z) - \log m(x_{ik'}, x_{jl}, z)) = 0. \end{aligned} \quad (19)$$

Define the discrete score function of $X_i$ as $g(x_{i1}, x_{ik'}, \gamma) = \log m(x_{i1}, \gamma) - \log m(x_{ik'}, \gamma)$, where $\gamma$ denotes other variables. Eq. 19 means $g(x_{i1}, x_{ik'}, \gamma)$ doe not take the value of $X_j$ as an argument when the LHS of Eq. 19 equals zero. As a result, Eq. 19 could be formulated as

$$\log m(x_{i1}, x_{j1}, z) = \log m(x_{ik'}, x_{j1}, z) + g(x_{i1}, x_{ik'}, \gamma). \quad (20)$$

Then by iterating all possible differences w.r.t. $X_i$, for all $k$ in $I_{x_i} = \{2, \ldots, M_i\}$, we have

$$\log m(x_{i1}, x_{j1}, z) = \log m(x_{ik}, x_{j1}, z) + g(x_{i1}, x_{ik}, \gamma). \quad (21)$$

By summation, we have

$$\begin{aligned} (N-1)\log m(x_{i1}, x_{j1}, z) &= \sum_{k \in I_{x_i}} (\log m(x_{ik}, x_{j1}, z) + g(x_{i1}, x_{ik}, \gamma)) \\ &= \sum_{k \in I_{x_i}} \log m(x_{ik}, x_{j1}, z) + \sum_{k \in I_{x_i}} g(x_{i1}, x_{ik}, \gamma), \end{aligned} \quad (22)$$

which implies that

$$\begin{aligned} M_i \log m(x_{i1}, x_{j1}, z) &= \sum_{k=1}^{M_i} \log m(x_{ik}, x_{j1}, z) + \sum_{k \in I_{x_i}} g(x_{i1}, x_{ik}, \gamma), \\ \log m(x_{i1}, x_{j1}, z) &= \frac{1}{M_i} \left( \sum_{k=1}^{M_i} \log m(x_{ik}, x_{j1}, z) + \sum_{k \in I_{x_i}} g(x_{i1}, x_{ik}, \gamma) \right). \end{aligned} \quad (23)$$

Because $\sum_{k=1}^{M_i} \log m(x_{ik}, x_{j1}, z)$ covers all possible values of $X_i$, this term does not depend on the specific value of $X_i$. Besides, the other term $\sum_{k \in I_{x_i}} g(x_{i1}, x_{ik}, \gamma)$ does not depend on $X_j$. It is worth noting that $X_{i1}$ could be any value of $X_i$ w.o.l.g.. Therefore, we could see that when the finite difference of the discrete score function of $X_i$ w.r.t. to $X_j$ equal to zero (after some aggregation of samples), $X_i \perp\!\!\!\perp X_j \mid \mathbf{Z}$.

**Necessary condition.** When $X_i \perp\!\!\!\perp X_j \mid \mathbf{Z}$, we could decompose $m(x_i, x_j, z)$ as $m_{X_i|\mathbf{Z}}(x_i \mid z)m_{X_j|\mathbf{Z}}(x_j \mid z)m_{\mathbf{Z}}(z)$. This implies that, for all $k$ in $\{2, \ldots, M_i\}$ and $l$ in $\{2, \ldots, M_j\}$, we have

$$
\begin{aligned}
&(\log m(x_{i1}, x_{j1}, z) - \log m(x_{ik}, x_{j1}, z)) \\
&\quad - (\log m(x_{i1}, x_{jl}, z) - \log m(x_{ik}, y_{jl}, z)) \\
&= \big(\log\big(m_{X_i|\mathbf{Z}}(x_{i1} \mid z)m_{X_j|\mathbf{Z}}(x_{j1} \mid z)m_{\mathbf{Z}}(z)\big) \\
&\qquad - \log\big(m_{X_i|\mathbf{Z}}(x_{ik} \mid z)m_{X_j|\mathbf{Z}}(x_{j1} \mid z)m_{\mathbf{Z}}(z)\big) \\
&\qquad - \big(\log\big(m_{X_i|\mathbf{Z}}(x_{i1} \mid z)m_{X_j|\mathbf{Z}}(x_{jl} \mid z)m_{\mathbf{Z}}(z)\big) \\
&\qquad\quad - \log\big(m_{X_i|\mathbf{Z}}(x_{ik} \mid z)m_{X_j|\mathbf{Z}}(x_{jl} \mid z)m_{\mathbf{Z}}(z)\big)\big) \\
&= \big(\big(\log m_{X_i|\mathbf{Z}}(x_{i1} \mid z) + \log m_{X_j|\mathbf{Z}}(x_{j1} \mid z) + \log m_{\mathbf{Z}}(z)\big) \\
&\qquad - \big(\log m_{X_i|\mathbf{Z}}(x_{ik} \mid z) + \log m_{X_j|\mathbf{Z}}(x_{j1} \mid z) + \log m_{\mathbf{Z}}(z)\big)\big) \\
&\qquad - \big(\big(\log m_{X_i|\mathbf{Z}}(x_{i1} \mid z) + \log m_{X_j|\mathbf{Z}}(x_{jl} \mid z) + \log m_{\mathbf{Z}}(z)\big) \\
&\qquad\quad - \big(\log m_{X_i|\mathbf{Z}}(x_{ik} \mid z) + \log m_{X_j|\mathbf{Z}}(x_{jl} \mid z) + \log m_{\mathbf{Z}}(z)\big)\big) \\
&= 0.
\end{aligned}
\tag{24}
$$

Therefore, when $X_i \perp\!\!\!\perp X_j \mid \mathbf{Z}$, the finite difference of the discrete score function of $X_i$ w.r.t. to $X_j$ equals zero.

The proof is complete. $\qquad\square$

## A.3 PROOF OF COROLLARY 2

**Corollary 2.** *Assume*

    i. $\mathbf{X} = (X_1, \ldots, X_d)$ *is a set of discrete variable.*

    ii. *The PMFs of $\mathbf{X}$ are strictly positive.*

    iii. *The characterization matrix $\Omega^{[d]}$ is defined according to Eq. (5).*

*Then for any $i \neq j$, $\Omega^{[d]}_{i,j} = 0$ implies $X_i \perp\!\!\!\perp X_j \mid X_{\mathbf{V}\setminus\{i,j\}}$.*

*Proof.* According to Eq. (5), we have

$$
\Omega^{[d]}_{i,j} := \mathbb{E}_{m_{\mathbf{X}}}\left[\sum_{k,l} f^{[d]}(x_{i1}, x_{ik}, x_{j1}, x_{jl}, z)^2\right],
\tag{25}
$$

where $f^{[d]}(x_{i1}, x_{ik}, x_{j1}, x_{jl}, z)$ denotes the LHS of Eq. (4), i.e.,

$$
\begin{aligned}
&f^{[d]}(x_{i1}, x_{ik}, x_{j1}, x_{jl}, z) \\
&= (\log m(x_{i1}, x_{j1}, z) - \log m(x_{ik}, x_{j1}, z)) \\
&\quad - (\log m(x_{i1}, x_{jl}, z) - \log m(x_{ik}, x_{jl}, z)).
\end{aligned}
\tag{26}
$$

Thus, if $\Omega^{[d]}_{i,j} = 0$, we must have

$$
\begin{aligned}
&(\log m(x_{i1}, x_{j1}, z) - \log m(x_{ik}, x_{j1}, z)) \\
&\quad - (\log m(x_{i1}, x_{jl}, z) - \log m(x_{ik}, x_{jl}, z)) = 0,
\end{aligned}
\tag{27}
$$

for all $k \in [M_i]$ and $l \in [M_j]$, where $M_i$ and $M_j$ denote the cardinalities of $X_i$ and $X_j$, respectively. Based on Theorem. 1, we have $X_i \perp\!\!\!\perp X_j \mid \mathbf{Z}$.

The proof is complete. $\qquad\square$

## A.4 PROOF OF THEOREM 2

**Theorem 2.** *Denote $\mathbf{V}$ as a set of mixed-type variables and $X_i, X_j \in \mathbf{V}$, where $X_i$ is discrete and $X_j$ is continuous. Let $\{x_{i1}, \ldots, x_{iM_i}\}$ be the support of variables $X_i$. For brevity, denote*

$\mathbf{V} \setminus \{X_i, X_j\}$ as $\mathbf{Z}$. Denote $z$ as any value(s) of $\mathbf{Z}$ and $x_j$ as any value of the continuous variable $X_j$. Then, $X_i \perp\!\!\!\perp X_j \mid \mathbf{Z}$ if and only if, for all $k \in [M_i]$ with $k \neq 1$, we have

$$\frac{\partial \log \left( p_{X_j, \mathbf{Z} \mid X_i}(x_j, z \mid x_{i1}) m_{X_i}(x_{i1}) \right)}{\partial x_j} - \frac{\partial \log \left( p_{X_j, \mathbf{Z} \mid X_i}(x_j, z \mid x_{ik}) m_{X_i}(x_{ik}) \right)}{\partial x_j} = 0. \tag{6}$$

*Proof.* **Sufficient condition.** Without loss of generality, let us consider three variables, i.e., $\{X_i, X_j, Z\}$:

$$\begin{aligned} x_i &\in \{x_{i1}, \ldots, x_{iM_i}\} \\ x_j &\in \mathbb{R}, \end{aligned} \tag{28}$$

where we set $X_i = x_i$ as the discrete variable and $X_j = x_j$ as the continuous variable w.l.o.g. Note that we do not constraint the type of $Z = z$ here but set $Z$ as continuous for brevity.

Consider the case that the finite difference of the score function of $X_j$ w.r.t. $X_i$ equals zero. Also, we define $p$ as the p.d.f. and $m$ as the p.m.f.. We first consider the difference between $x_{i1}$ and $x_{ik'}$.

$$\frac{\partial \log \left( p_{X_j, Z \mid X_i}(x_j, z \mid x_{i1}) m_{X_i}(x_{i1}) \right)}{\partial x_j} - \frac{\partial \log \left( p_{X_j, Z \mid X_i}(x_j, z \mid x_{ik'}) m_{X_i}(x_{ik'}) \right)}{\partial x_j} = 0. \tag{29}$$

By iterating all possible differences w.r.t. $x_i$, for all $k$ in $I_{x_i} = \{2, \ldots, M\}$, we have

$$\frac{\partial \log \left( p_{X_j, Z \mid X_i}(x_j, z \mid x_{i1}) m_{X_i}(x_{i1}) \right)}{\partial x_j} - \frac{\partial \log \left( p_{X_j, Z \mid X_i}(x_j, z \mid x_{ik}) m_{X_i}(x_{ik}) \right)}{\partial x_j} = 0, \tag{30}$$

which is equivalent to

$$\frac{\partial \log \left( p_{X_j, Z \mid X_i}(x_j, z \mid x_{i1}) m_{X_i}(x_{i1}) \right)}{\partial x_j} = \frac{\partial \log \left( p_{X_j, Z \mid X_i}(x_j, z \mid x_{ik}) m_{X_i}(x_{ik}) \right)}{\partial x_j}. \tag{31}$$

Then by integrating on both sides w.r.t. $X_j$, we have

$$\log \left( p_{X_j, Z \mid X_i}(x_j, z \mid x_{i1}) m_{X_i}(x_{i1}) \right) = \log \left( p_{X_j, Z \mid X_i}(x_j, z \mid x_{ik}) m_{X_i}(x_{ik}) \right) + C_k, \tag{32}$$

where $C_k$ is a constant. We then apply a summation as follows

$$\begin{aligned} &(M_i - 1) \log \left( p_{X_j, Z \mid X_i}(x_j, z \mid x_{i1}) m_{X_i}(x_{i1}) \right) \\ &= \sum_{k \in I_{x_i}} \left( \log \left( p_{X_j, Z \mid X_i}(x_j, z \mid x_{ik}) m_{X_i}(x_{ik}) \right) + C_k \right), \end{aligned} \tag{33}$$

which implies that

$$\begin{aligned} &\log \left( p_{X_j, Z \mid X_i}(x_j, z \mid x_{i1}) m_{X_i}(x_{i1}) \right) \\ &= \frac{1}{M_i} \left( \sum_{k=1}^{M_i} \log \left( p_{X_j, Z \mid X_i}(x_j, z \mid x_{ik}) m_{X_i}(x_{ik}) \right) + \sum_{k \in I_{x_i}} C_k \right). \end{aligned} \tag{34}$$

Because $\sum_{k=1}^{M_i} \log \left( p_{X_j, Z \mid X_i}(x_j, z \mid x_{ik}) m_{X_i}(x_{ik}) \right)$ covers all possible values of $k$, this term does not depend on the specific value of $X_i$. Besides, $C_k$ does not depend on $X_j$. Therefore, by iterating all possible differences of $X_i$, we could see that when the finite difference of the score function of $X_j$ w.r.t. $X_i$ equals zero (after some aggregation of samples), $X_i \perp\!\!\!\perp X_j \mid Z$.

It is noteworthy that another "symmetric" case, where the derivative of the discrete score function of $X_i$ w.r.t. $X_j$ equals zero, is as follows

$$\frac{\partial \left( \log \left( (p_{X_j, Z \mid X_i}(x_j, z \mid x_{i1}) m_{X_i}(x_{i1}) \right) - \log \left( p_{X_j, Z \mid X_i}(x_j, z \mid x_{ik}) m_{X_i}(x_{ik}) \right) \right) \right)}{\partial x_j} = 0, \tag{35}$$

which is equivalent to

$$\frac{\partial \log \left( p_{X_j, Z \mid X_i}(x_j, z \mid x_{i1}) m_{X_i}(x_{i1}) \right)}{\partial x_j} - \frac{\partial \log \left( p_{X_j, Z \mid X_i}(x_j, z \mid x_{ik}) m_{X_i}(x_{ik}) \right)}{\partial x_j} = 0. \tag{36}$$

Thus, we only need to consider Eq. 36, which is the case that the finite difference of the discrete score function of $X_i$ w.r.t. $X_j$ equals zero.

**Necessary condition.** When $X_i \perp\!\!\!\perp X_j \mid Z$, we could decompose $p_{X_j, Z \mid X_i}(x_j, z \mid x_{i1}) m_{X_i}(x_{i1})$ as $m_{X_i \mid Z}(x_{i1} \mid z) p_{X_j \mid Z}(x_j \mid z) p_Z(z)$. This implies that, for all $k$ in $\{2, \ldots, M_i\}$, we have

$$
\begin{aligned}
&\frac{\partial \log \left( p_{X_j, Z \mid X_i}(x_j, z \mid x_{i1}) m_{X_i}(x_{i1}) \right)}{\partial x_j} - \frac{\partial \log \left( p_{X_j, Z \mid X_i}(x_j, z \mid x_{ik}) m_{X_i}(x_{ik}) \right)}{\partial x_j} \\
=& \frac{\partial \log \left( m_{X_i \mid Z}(x_{i1} \mid z) p_{X_j \mid Z}(x_j \mid z) p_Z(z) \right)}{\partial x_j} \\
&- \frac{\partial \log \left( m_{X_i \mid Z}(x_{ik} \mid z) p_{X_j \mid Z}(x_j \mid z) p_Z(z) \right)}{\partial x_j} \\
=& \frac{\partial \left( \log m_{X_i \mid Z}(x_{i1} \mid z) + \log p_{X_j \mid Z}(x_j \mid z) + \log p_Z(z) \right)}{\partial x_j} \\
&- \frac{\partial \left( \log m_{X_i \mid Z}(x_{ik} \mid z) + \log p_{X_j \mid Z}(x_j \mid z) + \log p_Z(z) \right)}{\partial x_j} \\
=& \frac{\partial \log p_{X_j \mid Z}(x_j \mid z)}{\partial x_j} - \frac{\partial \log p_{X_j \mid Z}(x_j \mid z)}{\partial x_J} \\
=& 0.
\end{aligned}
\tag{37}
$$

Therefore, when $X_i \perp\!\!\!\perp X_j \mid Z$, the finite difference of the score function of $X_j$ w.r.t. to $X_i$ equal to zero.

The proof is complete. $\square$

## A.5  PROOF OF COROLLARY 3

**Corollary 3.** *Assume*

    *i.* $\mathbf{X} = (X_1, \ldots, X_d)$ *is a set of variables containing both continuous and discrete variables.*

    *ii. For continuous variables, the PDFs are strictly positive and smooth.*

    *iii. For discrete variables, the PMFs are strictly positive.*

    *iv. The characterization matrix $\Omega^{[m]}$ is defined according to Eq.* (7).

*Then for any $i \neq j$, $\Omega_{i,j}^{[m]} = 0$ implies $X_i \perp\!\!\!\perp X_j \mid \mathbf{X}_{\mathbf{V} \setminus \{i,j\}}$.*

*Proof.* According to Eq. (7), we have

$$
\Omega_{i,j}^{[m]} := \begin{cases}
\mathbb{E}_{\pi_{\mathbf{X}}} \left[ f_{i,j}^{[c]}(\mathbf{x})^2 \right] & \text{if } X_i \in \mathbf{X}_c, X_j \in \mathbf{X}_c \\
\mathbb{E}_{\pi_{\mathbf{X}}} \left[ \sum_{k,l} f^{[d]}(x_{i1}, x_{ik}, x_{j1}, x_{jl}, z)^2 \right] & \text{if } X_i \in \mathbf{X}_d, X_j \in \mathbf{X}_d \\
\mathbb{E}_{\pi_{\mathbf{X}}} \left[ \sum_k f^{[m]}(x_i, x_{j1}, x_{jk}, z)^2 \right] & \text{if } X_i \in \mathbf{X}_c, X_j \in \mathbf{X}_d \\
\mathbb{E}_{\pi_{\mathbf{X}}} \left[ \sum_k f^{[m]}(x_j, x_{i1}, x_{ik}, z)^2 \right] & \text{if } X_i \in \mathbf{X}_d, X_j \in \mathbf{X}_c,
\end{cases}
\tag{38}
$$

where $\mathbf{X}_c$ and $\mathbf{X}_d$ are the sets of continuous and discrete variables, respectively. We have already proved the first two cases (i.e., $\{X_i \in \mathbf{X}_c, X_j \in \mathbf{X}_c\}$ and $\{X_i \in \mathbf{X}_d, X_j \in \mathbf{X}_d\}$) in the proofs of Cor. 1 and Cor. 2, respectively. So here we will focus on the other two cases. We start from the third case, where $\{X_i \in \mathbf{X}_c, X_j \in \mathbf{X}_d\}$. We have

$$
\begin{aligned}
f^{[m]}(x_i, x_{j1}, x_{jk}, z) =& \frac{\partial \log \left( p_{X_i, \mathbf{Z} \mid X_j}(x_i, z \mid x_{j1}) m_{X_j}(x_{j1}) \right)}{\partial x_i} \\
&- \frac{\partial \log \left( p_{X_i, \mathbf{Z} \mid X_j}(x_i, z \mid x_{jk}) m_{X_j}(x_{jk}) \right)}{\partial x_i}.
\end{aligned}
\tag{39}
$$

Thus, if $\Omega_{i,j}^{[m]} = 0$ for $\{X_i \in \mathbf{X}_c, X_j \in \mathbf{X}_d\}$, we must have

$$\frac{\partial \log \left( p_{X_i, \mathbf{Z}|X_j}(x_i, z \mid x_{j1}) m_{X_j}(x_{j1}) \right)}{\partial x_i}$$
$$- \frac{\partial \log \left( p_{X_i, \mathbf{Z}|X_j}(x_i, z \mid x_{jk}) m_{X_j}(x_{jk}) \right)}{\partial x_i} = 0, \tag{40}$$

for all $k \in [M_j]$, where $M_j$ denotes the cardinality of $X_j$. Thus, according to Theorem 2, we have $X_i \perp\!\!\!\perp X_j \mid \mathbf{Z}$ if $\Omega_{i,j}^{[m]} = 0$ for $\{X_i \in \mathbf{X}_c, X_j \in \mathbf{X}_d\}$.

The similar derivation applies for the last case, where $\{X_i \in \mathbf{X}_d, X_j \in \mathbf{X}_c\}$.  $\square$

## A.6  PROOF OF COROLLARY 4

We first introduce the following lemmas and their proofs for completeness.

### A.6.1  PROOF OF LEMMA 1

**Lemma 1.** *[directly from Thm. 1 in (Hyvärinen & Dayan, 2005)] Assume*

  i. $\mathbf{X} = (X_1, \ldots, X_d)$ *is a set of continuous variables.*

  ii. *The data PDF $p_{\mathbf{X}}(\mathbf{x})$ is differentiable. The model PDF $(\mathbf{x}; \theta)$ is twice-differentiable. Both of them are strictly positive.*

  iii. *The expectations $E_{\mathbf{x}} \left\{ \| \log p(\mathbf{x}; \theta) \|^2 \right\}$ and $E_{\mathbf{x}} \left\{ \| \log p_{\mathbf{X}}(\mathbf{x}) \|^2 \right\}$ are finite for any $\theta$, and $p_{\mathbf{X}}(\mathbf{x}) \log p(\mathbf{x}; \theta)$ goes to zero for any $\theta$ when $\|\mathbf{x}\| \to \infty$.*

*Then Eq. (8) is equivalent to*

$$O_c(\theta) = \int_{\mathbf{x} \in \mathbb{R}^n} p_{\mathbf{X}}(\mathbf{x}) \sum_{i=1}^{d} \left[ \frac{1}{2} \| \nabla_{x_i} \log p(\mathbf{x}; \theta) \|^2 + H_{x_i}(\log p(\mathbf{x}; \theta)) \right] d\mathbf{x} + \rho_\lambda(\Omega^{[c]}). \tag{41}$$

*Proof.* Based on Eq. (8), we have

$$O_c(\theta) = \frac{1}{2} \int p_{\mathbf{X}}(\mathbf{x}) |\nabla_{\mathbf{x}} \log p(\mathbf{x}; \theta) - \nabla_{\mathbf{x}} \log p_{\mathbf{X}}(\mathbf{x})|^2 d\mathbf{x} + \rho_\lambda(\Omega^{[c]}), \tag{42}$$

where $\rho_\lambda(\cdot)$ denotes a sparsity penalty function and $\lambda$ is the penalty parameter. This is equivalent to

$$O_c(\theta) = \frac{1}{2} \int p_{\mathbf{X}}(\mathbf{x}) \left( \| \nabla_{\mathbf{x}} \log p(\mathbf{x}; \theta) \|^2 + \| \nabla_{\mathbf{x}} \log p_{\mathbf{X}}(\mathbf{x}) \|^2 \right.$$
$$\left. - 2 \left( \nabla_{\mathbf{x}} \log p_{\mathbf{X}}(\mathbf{x}) \right)^\top \left( \nabla_{\mathbf{x}} \log p(\mathbf{x}; \theta) \right) d\mathbf{x} + \rho_\lambda(\Omega^{[c]}). \tag{43}$$

We first consider the integral for the following part

$$- \int p_{\mathbf{X}}(\mathbf{x}) \left( \nabla_{\mathbf{x}} \log p_{\mathbf{X}}(\mathbf{x}) \right)^\top \left( \nabla_{\mathbf{x}} \log p(\mathbf{x}; \theta) \right) d\mathbf{x}, \tag{44}$$

by which we could obtain

$$- \sum_i \int p_{\mathbf{X}}(\mathbf{x}) \left( \nabla_{x_i} \log p_{\mathbf{X}}(\mathbf{x}) \right) \left( \nabla_{x_i} \log p(\mathbf{x}; \theta) \right) d\mathbf{x}$$

$$= - \sum_i \int \left( \nabla_{x_i} p_{\mathbf{X}}(\mathbf{x}) \right) \left( \nabla_{x_i} \log p(\mathbf{x}; \theta) \right) d\mathbf{x}$$

$$= - \sum_i \int \left[ \int \nabla_{x_i} p_{\mathbf{X}}(\mathbf{x}) \left( \nabla_{x_i} \log p(\mathbf{x}; \theta) \right) dx_1 \right] d(x_1, \ldots, x_d)$$

$$\overset{(\star)}{=} - \sum_i \int \left[ \lim_{a \to \infty, b \to -\infty} [p_{\mathbf{X}}(a, x_2, \ldots, x_d) \nabla_{x_i} \log p(a, x_2, \ldots, x_d, \theta) \right.$$
$$- p_{\mathbf{X}}(b, x_2, \ldots, x_n) \nabla_{x_i} \log p(b, x_2, \ldots, x_d, \theta)]$$
$$\left. - \int \frac{\partial^2 \log p_{\mathbf{X}}}{\partial x_i^2} p_{\mathbf{X}}(\mathbf{x}) dx_1 \right] d(x_2, \ldots, x_d), \tag{45}$$

where Eq. $(\star)$ is because if we assume $f$ and $g$ are both differential, we have

$$\frac{\partial f(\mathbf{x})g(\mathbf{x})}{\partial x_i} = f(\mathbf{x})\frac{\partial g(\mathbf{x})}{\partial x_1} + g(\mathbf{x})\frac{\partial f(\mathbf{x})}{\partial x_1}. \tag{46}$$

For $i \neq 1$, the cases follow similarly. Because we assume $p_{\mathbf{X}}(\mathbf{x}) \log p(\mathbf{x}; \theta)$ goes to zero for any $\theta$ when $\|\mathbf{x}\| \to \infty$, the limit is zero. Thus, we have proven that

$$-\sum_i \int p_{\mathbf{X}}(\mathbf{x})\nabla_{x_i} \log p_{\mathbf{X}}(\mathbf{x}) \, \nabla_{x_i} \log p(\mathbf{x}; \theta)) \, d\mathbf{x} = \sum_i \int \frac{\partial^2 \log p_{\mathbf{X}}}{\partial x_i{}^2} p_{\mathbf{X}}(\mathbf{x}) d\mathbf{x}, \tag{47}$$

By injecting it into Eq. (43), we obtain

$$O_c(\theta) = \int p_{\mathbf{X}}(\mathbf{x}) \Big[ \frac{1}{2}\|\nabla_{\mathbf{x}} \log p(\mathbf{x}; \theta)\|^2 + \frac{1}{2}\|\nabla_{\mathbf{x}} \log p_{\mathbf{X}}(\mathbf{x})\|^2$$
$$+ \operatorname{tr}\left(H_{\mathbf{x}}(\log p(\mathbf{x}; \theta))\right)] d\mathbf{x} + \rho_\lambda(\Omega^{[c]}). \tag{48}$$

Because $\frac{1}{2}\|\nabla_{\mathbf{x}} \log p_{\mathbf{X}}(\mathbf{x})\|^2$ does not depend on $\theta$, we could ignore it. Then we have

$$O_c(\theta) = \int \sum_{i=1}^d \left[ \frac{1}{2}\|\nabla_{x_i} \log p(\mathbf{x}; \theta)\|^2 + H_{x_i}(\log p(\mathbf{x}; \theta)) \right] d\mathbf{x} + \rho_\lambda(\Omega^{[c]}). \tag{49}$$

Thus, the proof is complete. $\qquad\square$

### A.6.2 PROOF OF LEMMA 2

**Lemma 2.** *[directly from (Lyu, 2012)] Assume*

    *i. $\mathbf{X} = (X_1, \ldots, X_d)$ is a set of discrete variables.*

    *ii. The data PMF $m_{\mathbf{X}}(\mathbf{x})$ and the model PMF $m(\mathbf{x}; \theta)$ are strictly positive.*

*Then Eq. (12) is equivalent to*

$$O_d(\theta) = \sum_{\mathbf{x}} m_{\mathbf{X}}(\mathbf{x}) \sum_{i=1}^d \left[ \left( \frac{\mathcal{M}_i(m(\mathbf{x}; \theta))}{m(\mathbf{x}; \theta)} \right)^2 - 2\mathcal{M}_i\left( \frac{\mathcal{M}_i(m(\mathbf{x}; \theta))}{m(\mathbf{x}; \theta)} \right) \right] + \rho_\lambda(\Omega^{[d]}). \tag{50}$$

*Proof.* Based on Eq. (12), we have

$$O_d(\theta) = \sum_{\mathbf{x}} m_{\mathbf{X}}(\mathbf{x}) \left\| \frac{\mathcal{M}(m(\mathbf{x}; \theta))}{m(\mathbf{x}; \theta)} - \frac{\mathcal{M}(m_{\mathbf{X}}(\mathbf{x}))}{m_{\mathbf{X}}(\mathbf{x})} \right\|^2 + \rho_\lambda(\Omega^{[d]}), \tag{51}$$

which implies

$$O_d(\theta) = \sum_{\mathbf{x}} m_{\mathbf{X}}(\mathbf{x}) \sum_{i=1}^d \left\| \frac{\mathcal{M}_i(m(\mathbf{x}; \theta))}{m(\mathbf{x}; \theta)} - \frac{\mathcal{M}_i(m_{\mathbf{X}}(\mathbf{x}))}{m_{\mathbf{X}}(\mathbf{x})} \right\|^2 + \rho_\lambda(\Omega^{[d]})$$
$$\overset{(\star)}{=} \sum_{\mathbf{x}} m_{\mathbf{X}}(\mathbf{x}) \sum_{i=1}^d \left[ \left( \frac{\mathcal{M}_i(m(\mathbf{x}; \theta))}{m(\mathbf{x}; \theta)} \right)^2 - 2\mathcal{M}_i\left( \frac{\mathcal{M}_i(m(\mathbf{x}; \theta))}{m(\mathbf{x}; \theta)} \right) \right] + \rho_\lambda(\Omega^{[d]}), \tag{52}$$

where Eq. $(\star)$ is due to the fact that $\left( \frac{\mathcal{M}_i(m_{\mathbf{X}}(\mathbf{x}))}{m_{\mathbf{X}}(\mathbf{x})} \right)^2$ does not take $\theta$ as an argument.

The proof is complete. $\qquad\square$

Then the corollary follows from these lemmas, which is included as follows.

**Corollary 4.** *Assume*

    *i. The data density $\pi_{\mathbf{X}}(\cdot)$ is equal to $\pi(\cdot; \theta^*)$ for some $\theta^*$.*

ii. *The data density $\pi_{\mathbf{X}}(\cdot)$ and model density $\pi(\cdot;\theta)$ are strictly positive. $\pi_{\mathbf{X}}(\cdot)$ and $\pi(\cdot;\theta)$ is differentiable and twice-differentiable, respectively, w.r.t. continuous variables. For some $\theta^*$, $\pi_{\mathbf{X}}(\cdot) = \pi(\cdot;\theta^*)$ and no other parameter value gives a density that is equal to $\pi(\cdot;\theta^*)$ almost everywhere.*

iii. *The expectations $E_{\pi_{\mathbf{X}}}\left[\|\log\pi(\mathbf{x};\theta)\|^2\right]$ and $E_{\pi_{\mathbf{X}}}\left[\|\log\pi_{\mathbf{X}}(\mathbf{x})\|^2\right]$ are finite for any $\theta$, and $\pi_{\mathbf{X}}(\mathbf{x})\log\pi(\mathbf{x};\theta)$ goes to zero for any $\theta$ when $\|\mathbf{x}\| \to \infty$.*

*Then $O'_m(\theta) = 0$ implies $\theta = \theta^*$.*

*Proof.* The $O'_m(\theta)$ is defined as follows

$$O'_m(\theta) = \mathbb{E}_{\pi_{\mathbf{X}}}\left[\sum_i s_i(\mathbf{x};\theta)\right], \tag{53}$$

where

$$s_i(\mathbf{x};\theta) \coloneqq \begin{cases} \frac{1}{2}\|\nabla_{x_i}\log\pi(\mathbf{x};\theta)\|^2 + H_{x_i}(\log\pi(\mathbf{x};\theta)) & X_i \in \mathbf{X}_c \\ \frac{1}{2}\left(\frac{\mathcal{M}_i(m(\mathbf{x};\theta))}{m(\mathbf{x};\theta)}\right)^2 - \mathcal{M}_i\left(\frac{\mathcal{M}_i(m(\mathbf{x};\theta))}{m(\mathbf{x};\theta)}\right) & X_i \in \mathbf{X}_d, \end{cases} \tag{54}$$

where the probability function $\pi$ is strictly positive. According to Lemma 1 and Lemma 2, both cases of $s_i(\cdot;\theta)$ in $O'_m(\theta)$ are equivalent to $\frac{1}{2}\mathbb{E}_\pi\left[\left\|\frac{g(\pi(\cdot;\theta))}{\pi(\cdot;\theta)} - \frac{g(\pi(\cdot))}{\pi(\cdot)}\right\|^2\right]$, where $g$ denotes the gradient operator for continuous variables or the marginalization operator for discrete variables. If $O'_m(\theta) = 0$, $s_i(x;\theta)$ must equal to zero for any $i$. Because of Brook's Lemma (Brook, 1964), which is also included in Lyu (2012) as Lemma 3, the marginalization operator $\mathcal{M}$ is complete (Defn. 1). Thus, for the discrete variables, we could replace the gradient in the continuous score function with the marginalization operator while preserving local consistency as that for the continuous variables, which is shown by Theorem 2 in Hyvärinen & Dayan (2005). $\qquad\square$

# B  EXPERIMENTS

## B.1  GENERATING PROCESS FOR MIXED-TYPE DATA

For the mixed-type case, we simulate data with the process described in Andrews et al. (2018), of which the details are included here for completeness. After generating a random decomposable DAG, we first assign a data type (continuous or discrete) to each variable with equal probability. For variables without parents in the ground-truth graph, we sample their values from Gaussian and Multinomial distributions, respectively. Then for each continuous variable, we create a temporary discretized version by applying equal frequency binning. The number of bins is uniformly chosen between and including 2 and 5. The cardinality of each discrete variable is uniformly chosen between and including 2 and 4. The randomly generated decomposable DAGs are moralized to obtain the ground-truth Markov network structures.

Next, for variables with parents in the ground-truth graph, we sample the values of them as follows. For continuous variables, we first adopt partitioning according to its discrete variables. Then the values of these continuous variables are generated by randomly parameterizing the coefficients of a regression for each partition. For discrete variables, we generate the values of them by randomly parameterizing Multinomial distributions of the variables of the target variable and its discrete parents (temporary or not). After the simulation, all temporary discretized variables are removed.

## B.2  INFLUENCE OF THE SAMPLE SIZE

In this section, we report additional experimental results with a larger sample size. We conduct experiments for all settings (continuous, discrete, and mixed-type) with different numbers of variables ($d \in \{4, 6, \dots, 20\}$) and 10000 samples. The results are summarized in Fig. 4, Fig. 5, and Fig. 6. One can observe that both KCI and GS fail in all settings, indicating that they cannot scale well with large sample sizes. It is because the complexities of KCI and GS grow cubically in the number of samples, which is one of the motivations for the development of our method. Besides, SING cannot

scale with more than 6 variables because of OOM. At the same time, our method works well across all datasets without any scalability issues. Together with the better performance illustrated in Sec. 4 (note that one can even go beyond 5000 variables, e.g., it takes 7725 seconds for 10,000 variables in our setting), we believe the potential of our method is not only theoretically exciting but also empirically clear in both consistency and scalability.

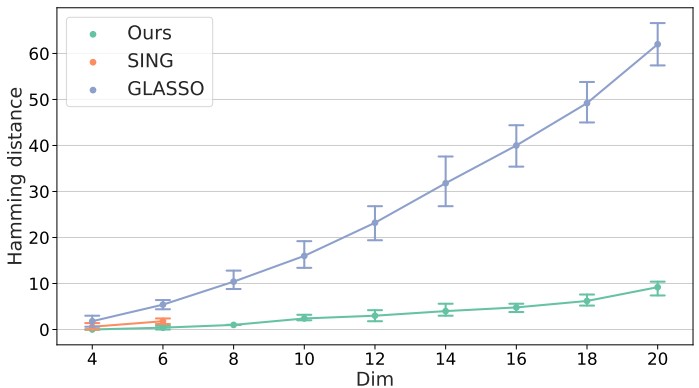

Figure 4: Hamming distances for continuous data, $n = 10000$.

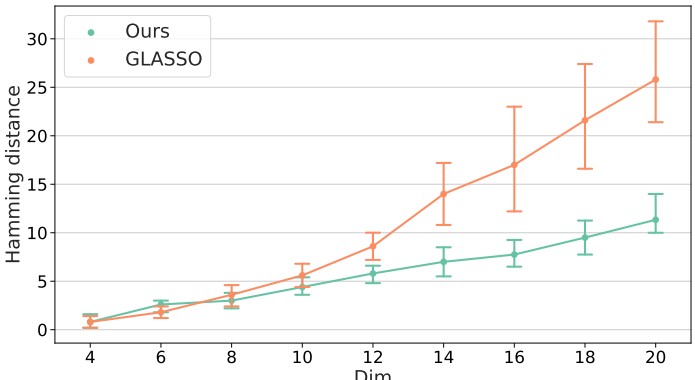

Figure 5: Hamming distances for discrete data, $n = 10000$.

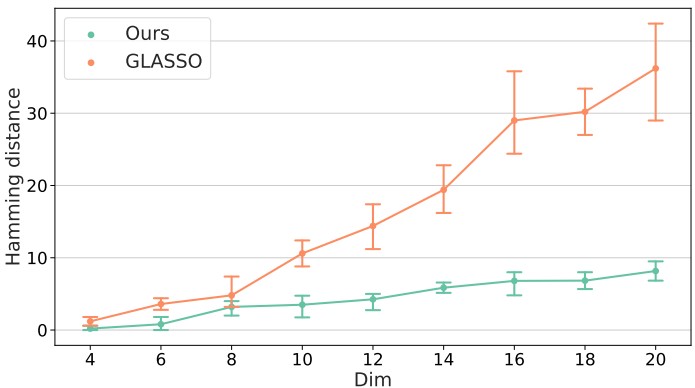

Figure 6: Hamming distances for mixed-type data, $n = 10000$.

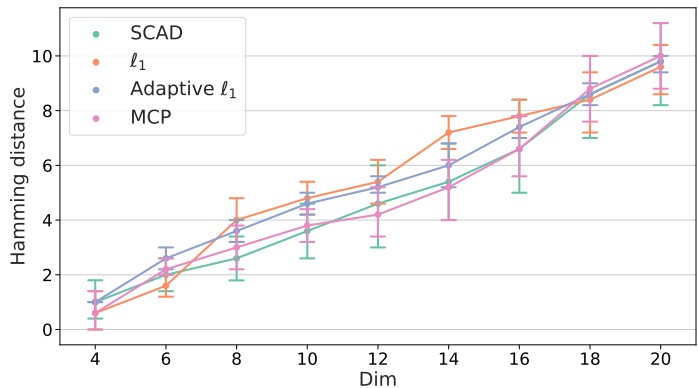

Figure 7: Hamming distances w.r.t. different sparsity penalty functions and numbers of variables.

B.3 INFLUENCE OF DIFFERENT PENALTY FUNCTIONS

To explore the effect of different regularization functions, we compare the results of our method with different sparsity penalties, which are shown in Fig. 7. The experiments are conducted on Butterfly distributions with the number of continuous variables ranging from 4 to 20 and a sample size of 1000. One could observe that SCAD and MCP outperform other penalties, while SCAD performs slightly better than MCP in general. Adaptive $\ell_1$ (Zou, 2006) also illustrates its advantage compared to the original $\ell_1$ penalty. This suggests the importance of appropriate penalty functions.

## C  DISCUSSION

### C.1  TOWARDS NONPARAMETRIC CAUSAL DISCOVERY

In this section, we briefly discuss the implication of our proposed Markov network estimation method in causal discovery, of which the goal is to learn graphical models with causal interpretations.

The major classes of approaches for causal discovery are constraint-based approaches that utilize conditional independence tests and score-based approaches that optimize a specific score function. Among them, PC (Spirtes & Glymour, 1991) with kernel-based conditional independence test (Zhang et al., 2012) and GES (Chickering, 2002) with generalized score (Huang et al., 2018) are able to handle nonparametric cases with assumptions such as causal sufficiency. Both of these approaches rely on kernel methods whose computational complexity is cubic w.r.t. the number of samples. Therefore, the running time could be long if the sample size is large. Furthermore, when the number of variables is large, the search procedure may involve computing the kernel-based conditional independence test or score function many times, which therefore may also increase the running time.

As shown by Loh & Bühlmann (2014); Ng et al. (2021) in the linear Gaussian case, the Markov network (i.e., the support of the inverse covariance matrix of the distribution) is guaranteed to be the super-structure of the ground truth directed acyclic graph (DAG) under a specific type of faithfulness assumption. That is, the super-structure contains all edges of the true DAG. Using this idea, they showed that the Markov network may be used to restrict the search space of score-based approaches for causal discovery, which improves the scalability. Their works focus only on the linear case and adopt classical methods like graphical Lasso (Friedman et al., 2008) to estimate the Markov network. In this work, the nonparametric Markov network estimated by our proposed procedure could potentially be used as a super-structure to restrict the search space for nonparametric causal discovery methods, i.e., (kernel-based) PC and GES. Similar to (Loh & Bühlmann, 2014; Ng et al., 2021), this may help reduce the running time and improve the scalability of these methods.

