# OpenReview forum: "Generalized Precision Matrix for Scalable Estimation of Nonparametric Markov Networks"
_ICLR.cc/2023/Conference — ICLR 2023 poster_

### Official Review · Reviewer_ircN · 2022-10-20

**Confidence:** 1
**Correctness:** 3
**Technical Novelty And Significance:** 3
**Empirical Novelty And Significance:** 3
**Recommendation:** 6

**Clarity, Quality, Novelty And Reproducibility:**

Overall, the paper is well written and the most technical parts are easy-to-follow.




**Strength And Weaknesses:**

Strength ：
It is the first attempt to estimate Markov networks for all data types (i.e.,continuous, discrete, and mixed-type)
The esitmation method is scalable ( up to 5000 nodes within one hour on CPUs )

Weaknesses ：
- There's no experiments on real-world examples except only some comparisons are conducted on synthetic examples.

Questions:
-  why do not consider comparing the method of [1] in your final experiments, for mixed-type data ?

[1] Liu, H., La erty, J. D., and Wasserman, L. (2009). The nonparanormal: Semiparametric estimation of high dimensional undirected graphs. The Journal of Machine Learning Research, 10:2295–2328.

**Summary Of The Paper:**

The paper proposes a scalable estimation method for non-parametric Markov network structures, using regularized score matching.
They first introduce necessary and sufficient conditions of conditional independence between variables in general distributions for all data types (i.e., continuous, discrete, and mixed-type) without specific assumptions on functional relations among variables.
They also introduce appropriate penalties on the characterization matrix, to promote constantly sparse entries for
stable estimation.

**Summary Of The Review:**

This paper presents a scalable estimation method for non-parametric Markov network structures.
It addresses the limitations of previous related methods in handling all data types (i.e., continuous, discrete, and mixed-type).
To me, it should be accepted based on its high quality and novelty.

---

> ### Author Response · Authors · 2022-11-15
> **We are very grateful for your time, encouraging comments, and positive feedback.**
>
> We are very grateful for your time, encouraging comments, and positive feedback. Please find the response to your questions below.
>
> **Q1:** Why do not consider comparing the method of [1] in the final experiments, for mixed-type data?
>
> **A1:** Thanks so much for the great suggestion. **We have implemented it and included the new comparison results in Section 4.** We observe that its performance (NPN in Figure  1 and Figure 2) is similar to (or slightly worse) than that of GLASSO. We conjecture that it may be due to its hypothesis of the nonparanormal transformation, which does not hold in the considered setting (general distributions with mixed-type data).
>
> Besides, in order to strengthen the **evaluation** of our method in **more complex scenarios**, we have **re-run all experiments on a set of new datasets**, where the Markov network structures are randomly generated. The settings and results have been included in the updated manuscripts (Section 4 and Appendix B). From these new results (Figure 2), we can observe a similar pattern that the advantages of our approach over baselines in terms of structure recovery are more obvious in complex scenarios (distributions from random graphs) than in rather simple ones (Butterfly distributions). Together with the much better scalability results (Table 1 and Figure 3), we believe the potential of our method is both theoretically exciting and empirically clear.
>
>
> Again, thanks for the appreciation of our work. We are really encouraged by it.
>
> ----------------------------------
>
> **References:**
>
> [1] Liu, Han, John Lafferty, and Larry Wasserman. "The nonparanormal: Semiparametric estimation of high dimensional undirected graphs." Journal of Machine Learning Research 10.10 (2009).

---

> > ### Author Response · Authors · 2022-11-18
> > **Please kindly let us know if you would like to suggest any further changes**
> >
> > Thanks so much for your constructive feedback. Besides the newly added baseline, please kindly let us know if any further updates on the manuscript are needed since the manuscript updating deadline is approaching.

---

### Official Review · Reviewer_XHnf · 2022-10-21

**Confidence:** 3
**Correctness:** 4
**Technical Novelty And Significance:** 3
**Empirical Novelty And Significance:** 3
**Recommendation:** 8

**Clarity, Quality, Novelty And Reproducibility:**

The theoretical part of the paper is written in a clear way and contains novel elements such as a characterisation of conditional independence for discrete and mixed type variables.
Moreover a new algorithm which is able to infer a Markov network from (mixed type) data is introduced.
The novelty of an algorithm that infers the MRF from mixed type data (as opposed to only continuous or discrete data) is unclear.  This is due to the fact that, for mixed type data, the authors benchmark their algorithm against two other algorithms.
In the current state not enough details about the numerical experiments are provided for them to be reproducible.
The true potential of the algorithm is at this point unclear, due to the very simple setting in which the authors test it.

**Strength And Weaknesses:**


Strengths
1. Sections 1 and 2 are well written and clear. Same for section 3 except for some minor problems (see below).
2. The characterization of conditional independence for discrete and mixed type variables in section is novel and has potential to be used also in different algorithms (not necessarily based on score matching)
3. Potentially much faster method of discovering conditional independence relations (but must be verified with further numerical experiments) compared to what exists in the literature.

Points of improvement
1. In page 3 line 6. It is unclear what it means that $\Omega$ is differentiable (i.e. differentiable with respect to what?)
2. In equation 9 should the integral be over $\mathbb R^d$ ?
3. The way to recover the matrix $\Omega$ once the score matching has been run is not immediate to see. Some comments about this are necessary (for example the average over $p$ in (3) is over $p_x(x)$ or over the fitted model $p(x|\theta)$? )
4. Typo regualrized  pag 9.
5. The novelty of an algorithm that infers Markov networks from mixed type data is unclear. One is in fact confused when, in the setting of mixed variables, the algorithm developed in this paper is compared with two other algorithms: KCI and GS. To make a stronger point about the novelty of the method one should comment on these two algorithms and, if relevant, mention them among the previous works.
6. For clarity I would suggest specifying that in a practical setting $p_x(x)$ is the the empirical data distribution given by $n$ samples, sampled from the data distribution.
7. The Markov Random Field considered in the experiments (Butterfly model) is very simple: the associated MRF is just a collection of r disconnected pairs of variables. This makes it hard to evaluate the potential of the method on more complex random fields. I would suggest using something like a random graph as MRF (for example following what is done in [Zhang et al. 2012])
8. Several details are missing in the numerical experiments. This information is critical to understand how the results were obtained and to appreciate their significance:

    (a) What are the parameters \theta used in the experiments? In other words what is the parametric family that is being fitted to the butterfly model?

    (b) How is the loss optimized? (e.g. gradient based methods,…)

    (c) Is the optimization guaranteed to reach the global minimum? Is the problem convex?

    (d) In the context of mixed type data the meaning of «mix the two types of pairs for the mixed type case» as a way of specifying how the data were generated is not clear. The description of the generative process for discrete data should also be made more precise.

    (e) When saying «we also conduct experiments on large graphs» all the details about these experiments must be provided.



**Summary Of The Paper:**

The paper introduces a novel technique to estimate Markov networks from data, that is, discovering conditional independence relations between variables. While previous approaches could only be applied to either continuous or discrete random variables, the paper presents a framework which applies to both and to mixed type data. The resulting algorithm is also faster compared to those in the literature. The approach used is based on regularized score matching. In the case of continuous data the paper uses a method already established in the literature, while for discrete type data the method itself presents some novel aspects. Some numerical experiments are presented to validate the claim of better scalability.

**Summary Of The Review:**

The paper contains some interesting results about the characterisation of conditional independence. Moreover a new algorithm is proposed to infer Markov networks from data. While the theoretical results seem sound, it is very difficult to evaluate the algorithm due to several missing details. A revision of the experimental part is needed to accept the paper.

In the rebuttal the authors have satisfactorily addressed the totality of the points I raised, for this reason I recommend that the paper be accepted.

---

> ### Author Response · Authors · 2022-11-15
> **We greatly appreciate the reviewer’s time, encouraging comments, and constructive suggestions (1/2).**
>
> We greatly appreciate the reviewer’s time, encouraging comments, and constructive suggestions. With the help of such valuable feedback, we believe that our manuscript could be improved a lot. We have updated **a new set of experiments** for all considered evaluations, **a new section including detailed experimental settings**, and **new results**. The manuscripts, especially the experimental part, have been thoroughly modified in both contents and structures according to all these suggestions. Please find the point-by-point responses below.
>
>
> **Q1:** It is unclear what $\Omega$ is differentiable with respect to.
>
>
> **A1:** Thanks for pointing this out. $\Omega$ is differentiable with respect to $\mathbf{X}$. We are sorry for the potential confusion and have updated the manuscript.
>
>
> **Q2:** Typos in Equation 9 and page 9.
>
>
> **A2:** Thanks for the reminder. We have fixed them in the updated manuscript.
>
>
> **Q3:** Some comments about the way to recover the matrix $\Omega$ once the score matching has been run are necessary.
>
>
> **A3:** We appreciate the reviewer for raising this point. Basically, the characterization matrix $\Omega$ is obtained along with the regularized score matching process as part of the regularization term. After estimation, the expectation (e.g., Equation 3) is computed over the fitted model $p(\mathbf{x};\theta)$. We have added the corresponding details in the updated manuscript.
>
>
> **Q4:** It will make the novelty of the method for mixed-type data clearer if there are more comments on KCI and GS.
>
>
> **A4:** We are grateful for this essential suggestion. Besides the current discussion in the experiment section, we have updated the manuscript to include more discussion of these two methods among the previous works in the introduction section. Briefly, although KCI and GS could handle the general setting, their complexities scale cubically in the number of samples, which makes the computation challenging. The scalability issue can also be observed in our experimental results.
>
>
> **Q5:** For clarity, it is better to specify that, in a practical setting, $p_{\mathbf{X}}(\mathbf{x})$ is the empirical PDF.
>
>
> **A5:** Thanks so much for this great suggestion. We have updated the manuscript accordingly.
>
>
> **Q6:** More complex structures for evaluations (e.g., random graphs).
>
>
> **A6:** Thanks for raising this constructive suggestion, We fully agree with you that experiments on more complex graphs can be helpful to better evaluate the proposed method. According to your suggestions, we have conducted a new set of experiments with random graphs for all considered evaluations, of which the detailed settings, results, and observations are included in the new experiment section and Appendix (Section 4 and Appendix B).
>
>
> From these new results on random graphs (Figure 2), we can observe a similar pattern that the advantages of our approach over baselines in terms of structure recovery are more obvious in complex scenarios (distributions from random graphs) than in rather simple ones (Butterfly distributions). Together with the much better scalability results (Table 1 and Figure 3), we believe the potential of our method is both theoretically exciting and empirically clear.

---

> > ### Author Response · Authors · 2022-11-15
> > **We greatly appreciate the reviewer’s time, encouraging comments, and constructive suggestions (2/2).**
> >
> > **Q7:** Several details are missing in the numerical experiments.
> >
> >
> > **A7:** We are grateful for all those great questions regarding the details of the experimental settings. We believe they are essential for a better evaluation and have updated them in the manuscript. Please see the point-by-point responses below:
> >
> >
> > -  **Q7(a):** What are the parameters $\theta$ used in the experiments?
> >
> >
> >
> >    **A7(a):** Thanks for the great question. We used the deep kernel exponential family (DKEF) in the experiment. Generally, other deep estimation models could also be used depending on various contexts such as computational resources and experimental settings. We have updated it in the current manuscript.
> >
> >
> >
> > - **Q7(b):** How is the loss optimized?
> >
> >     **A7(b):** Thanks for the reminder. The loss is optimized by gradient-based optimization with the Adam optimizer.
> >
> >
> > - **Q7(c):** Is the optimization guaranteed to reach the global optimum? Is the problem convex?
> >
> >     **A7(c):** We appreciate this great question. No, it is not guaranteed to reach the global optimum and the problem is nonconvex. The estimation process may often reach local optimum as the overall optimization problem is non-convex, as is typical for methods involving deep learning.. Moreover, the smoothly clipped absolute deviation (SCAD) penalty we adopted is also nonconvex. Nevertheless, the empirical results in Section X show that the local solutions still give rise to competitive performance in practice.
> >
> >
> > - **Q7(d):** The details of the data generating process should be richer and clearer.
> >
> >     **A7(d):** Thanks for the valuable suggestion. We have updated the experiment section and included a detailed description of the data generating process in light of your suggestion.
> >
> >
> > - **Q7(e):** The details of the experimental settings on large graphs should be richer and clearer.
> >
> >     **A7(e):** We sincerely appreciate the great suggestion. The corresponding details have been added to the updated manuscript.
> >
> >
> > Finally, we would like to thank you again for the time devoted and all these excellent comments, which are essential for the improvement of the manuscript. We especially appreciate your suggestions regarding how to make the evaluation stronger. Accordingly, we have thoroughly modified the experimental part, including sets of new experiments on more complex datasets. We hope our response and the updated manuscript could properly address your comments. Please kindly let us know if there are any remaining questions.

---

> > > ### Author Response · Authors · 2022-11-15
> > > **Thanks so much for checking the response and updating your recommendation**
> > >
> > > We are glad that our response and the updated manuscript have addressed your concerns. Thanks again for your valuable suggestions!

---

### Official Review · Reviewer_RUet · 2022-10-24

**Confidence:** 4
**Correctness:** 3
**Technical Novelty And Significance:** 2
**Empirical Novelty And Significance:** 2
**Recommendation:** 6

**Clarity, Quality, Novelty And Reproducibility:**

The review is organised according to the sections in the paper.

# Abstract

- Type, first line, should be: _A_ Markov network characterises...

# Introduction

- A Markov network does not have to be the same as an undirected graphical model - if the latter has the Markov property then yes they are the same thing, but if it does not, it is a different model altogether. Fix this, this is currently incorrect.
- The last sentence of your first paragraph is very unambiguous, you say "However, methods for Gaussian graphical models might fail to correctly capture dependencies between variables deviating from Gaussian or including nonlinearity" - what do you mean by 'might', in what cases does this happen? This is too imprecise. Give examples of when it fails and when it doesn't and why.
- Spell out what "SING" stands for instead of just stating the acronym - we as readers do not yet know what it stands for even if you do.

# GENERALIZED CHARACTERIZATION OF CONDITIONAL INDEPENDENCE

- It will probably help if you, in the second paragraph, talk about 'forks' and 'colliders' etc. to let the reader know the graphical topological structure you are referring to (in addition to your present discussion on conditional independence).
- I realise you are constrained for space, but it would be helpful if you could put (a) to (f) inside an itemized list to aid the exposure of your procedure. It is very dense at the moment and does not flow well.
- Notation: your notation is incorrect for $\mathbf{V} \setminus (I,j)$ it should be $\mathbf{V} \setminus \{I,j\}$ since you are removing the _set_ $\{i,j\}$.
- You variously refer to $\Omega$ as the characterisation as well as the matrix - which is to be? Stick to one, you are being inconsistent.
- The notation for equation 1 is _very_ busy and cluttered. You do not need that level of detail to make your point about the PDF: it is a fairly common idea and most will have seen it before.
- I do not understand where equation 2 comes from, explain that better alongside your smoothness assumption - right now you're just saying that $p_X$ is smooth. A smooth function is a function that has continuous derivatives up to some desired order over some domain. Perhaps you can be a bit more precise than just saying 'smooth'.
- Ok. You cannot just launch into the paragraph that follows equation 2. That will not doe. What is $g$, what is $h$ what is $n$ - where do they come from? Why do they have that structure? I understand that it is a result from Lauritzen, 1996, but it is not enough to just state without an explaining narrative to go along with it. Again, your readers do not know as much as you do, you have to explain basic things to us (like the aforementioned) and not assume that this is a known result.
- What is $c$ in equation 3? Why is it inside square braces (or should that be an Iversion bracket)? Which $p$ are we discussing here in the subscript of the expectation?
- "the gradient is undefined" - you make it sound like someone made a mistake in this case; the gradient doesn't exist at all.
- The last line of corollary 2, why are you using notation $\textbf{x}_i$ instead of $X_i$ and so on? I do not follow.
- I take it that you mean that $d$ in equation 5 is to stand for "discrete" and presumably too "c" for continuous? Just state that somewhere and further explain why these basic labels are inside braces.

# SCALABLE ESTIMATION WITH REGULARIZED SCORE MATCHING

- Why is the limit 5000 nodes? What happens if we use 10,000 nodes?
- How strong is the assumption of degeneracy?
- What is the domain of the penalty parameters, I assume [0,1]?
- You should perhaps be a bit less informal that to call something a "trick" and use less colloquial terms to refer to results from previous works.
- I am on page six and I still don't know what 'SING' stands for.
- Where does equation 10 come from? You say that it can be reformulated, but how? Through what means?
- I am not sure what you mean by saying that $\mathcal{L}$ is '"complete"' (your quote, not mine) - is the implication that it doesn't have to be?
- I find your notation a bit questionable, where you have used $\tilde$.

# TOWARDS NONPARAMETRIC CAUSAL DISCOVERY

- In your penultimate sentence of this section you say "our proposed procedure could potentially be used" - do you actually go on and use it for this?
- In your final sentence you say: "this may help reduce the running time and improve the scalability of these methods" - well, does it?

Implication alone is not a very powerful device if you do not demonstrate said implication in the paper. Certainly, many methods and ideas have lots of implications, but if they are not empirically or otherwise shown, they hold little value beyond the idea itself. Given that ML is a rather practical field, this is not very useful.

# EXPERIMENTS

- Finally, we are told what 'SING' stands for on page 8.
- If I have read your results correctly, the following is true: 1(a) GS wins on all dimensions; 1(b) it is unclear if GS or your method is better in all dimensions; 1(c) GS wins again 1(d): yours wins by a mile 1(e) you win against the only other method you are comparing against, a method which is only efficient on Gaussian distributions. Same with 1(f).
- I would recommend you scale your results with the figures in table 1 - this way your method will look far more impressive. At the moment, it looks like you are 'beaten' in half of your own experiments.


**Strength And Weaknesses:**

# Strengths

- The speed of the algorithm is very impressive and exciting. Further details need to be provided of course, of the experimental setup but the experiments are encouraging indeed.
- There is good mathematical rigour throughout.


# Weaknesses

- The sheer amount of results which are in the appendix, I find problematic. I understand, as ever, that there is not enough space but proofs are not trivial, they are results. At the very least, a sketch of the proof should be included in the body of the paper. The authors will be aware that reviewers are exceedingly busy and many of us do not have time to read all the appendices that come with the papers we are asked to review. You even have a lemma in the appendix. This venue may be inappropriate for such a large body of work, and a journal may be more suitable.
- There is a common theme where the authors like to spell out the acronym or some relevant method/algorithm/idea but not actually telling the reader what it stands for which is very bad practice.
- The English is lacking in places, prepositions not there and general sentence construction is off.

**Summary Of The Paper:**

The authors introduce an algorithm capable of learning very general Markov network structures, from general data-types including continuous, discrete and mixed.

**Summary Of The Review:**

There are many parts that I find questionable and I did not check the maths in detail (time constraints). I am willing to update my score should my concerns be adequately addressed. But as it stands this is a dense paper which lacks an adequate narrative to make it legible let alone understandable. There are too many unsupported claims, too many undefined acronyms and missing proofs (or even sketches) to make this (as it stands) a good paper.

---

> ### Author Response · Authors · 2022-11-15
> **We appreciate the reviewer for the time dedicated, constructive suggestions, and encouraging feedback (1/4).**
>
> We appreciate the reviewer for the time dedicated to reviewing our paper, constructive suggestions, and encouraging feedback. We have carefully modified **the structure and narrative of the manuscript** and added **several sets of new experiments** according to your detailed suggestions. We have put much effort into addressing all your concerns. Please find the responses to all your comments below.
>
> **Q1:** There is a sheer amount of proof in the appendix. Although the space is limited for such a large body of work, at the very least, a sketch of the proof should be included in the main part of the paper.
>
> **A1:** Thanks for your very helpful comment and being understanding. In light of your great suggestions, we have added proof sketches for theorems in the updated manuscript and modified the overall structure accordingly for a better flow. (Please also note that the discussion on nonparametric causal discovery was moved to the appendix instead.)
>
> **Q2:** The meaning of SING is missing until page 8.
>
> **A2:** We apologize for the confusion this may have caused in reading. We have fixed it and re-checked the manuscript thoroughly.
>
> **Q3:** Language issues in some places, including, for example, the typo of missing ‘a’ before ‘Markov network’.
>
> **A3:** Thanks for noticing it. We have corrected it accordingly.
>
> **Q4:** The sentence ‘Markov networks (also known as undirected graphical models or Markov random fields) …’ is not accurate. A Markov network does not have to be the same as an undirected graphical model - if the latter has the Markov property then yes they are the same thing, but if it does not, it is a different model altogether.
>
> **A4:** Thanks for pointing out the difference in the definitions of these two terms. We have fixed it by removing the term ‘undirected graphical models’ there.
>
> **Q5:** The sentence, "However, methods for Gaussian graphical models might fail to correctly capture dependencies between variables deviating from Gaussian or including nonlinearity", is ambiguous due to the word ‘might’.
>
> **A5:** Thanks for your nice suggestion. We have removed the word ‘might’ to make it more concrete. At the same time, we would also like to illustrate more details regarding applying Gaussian models on non-Gaussian data.
> - From a theoretical perspective, the theoretical result of model selection consistency for a Gaussian graphical model does not hold when the assumptions (e.g., Gaussianity or sub-Gaussianity) are not satisfied [1, 2]. For instance, in a general non-Gaussian setting, the relationship between entries of the inverse covariance matrix and the dependencies is unknown.
> - On the pragmatic side, it fails to correctly learn the structure in various empirical evaluations. For example, [3] applied Gaussian graphical models on non-Gaussian data and concluded quantitatively that it would not recover the correct graph.
>
> **Q6:** It will probably help if the graphical topological structures that the authors are referring to could be mentioned (in addition to the present discussion on conditional independence).
>
> **A6:** Thanks so much for this great point. We have added examples including ‘colliders’ to better illustrate the conditional independence and the definition of Markov property.
>
> **Q7:** It would be helpful if the authors could put (a) to (f) inside an itemized list to aid the exposure of the procedure.
>
> **A7:** We totally agree with you and appreciate your understanding about the space limitation. We have constructed them inside an itemized list in the updated manuscript for a better flow.

---

> > ### Author Response · Authors · 2022-11-15
> > **We appreciate the reviewer for the time dedicated, constructive suggestions, and encouraging feedback (2/4).**
> >
> > **Q8:** Comments regarding notation. These include **points 3, 4, 5, 6, 7, 10, and 11** in the review section “**Generalized characterization of conditional independence**” and **point 8** in the review section “**Scalable estimation with regularized score matching**”.
> >
> > **A8:** We are more than grateful for all those points that could definitely make the manuscript clearer. Thanks so much for the time devoted to reviewing the manuscript thoroughly and carefully. We also would like to apologize for any inconvenience caused by these notation issues/typos. We have made the following changes/clarification in light of your suggestions.
> >
> > 1. Thanks so much. We have changed it to $\mathbf{X}_{\mathbf{V} \backslash \{i, j \} }$  (with curly braces).
> >
> > 2. $\Omega$ is the characterization of conditional dependencies in a matrix form. Each entry in the matrix $\Omega$ (e.g., $\Omega_\{i, j\}$) corresponds to the characterization of conditional independence between variables $X_i$ and $X_j$ given the remaining variables. Thus, we denote $\Omega$ as the characterization matrix. We have made sure that it is consistent in the updated manuscript.
> >
> > 3. The notation in Equation 1 has been simplified for clarity.
> >
> > 4. The notation of the paragraph following Equation 2 has been clarified.
> >
> > 5. $c$ in Equation 3, as well as $d$ and $m$ in other equations, have been clarified as the indicators for continuous, discrete, and mixed-typed data, respectively. These labels as superscripts are inside braces to distinguish them from the power.
> >
> > 6. $p$ in the subscript of the expectation in Equation 3 has been changed to $p_{\mathbf{X}}$.
> >
> > 7. $\mathbf{x}_i$ in Corollary 2 has been corrected to $X_i$. Similar changes have been made to other terms in that sentence.
> >
> > 8. Thanks so much for the great question. We agree that $\sim$ might not be the optimal notation. We adopted it mainly in order to keep it consistent with the original paper [9]. We have adjusted the way it looks in the updated manuscript to make it more compact.
> >
> > Again, we sincerely thank you for all of those detailed comments regarding the notation. Thank you for so much time and effort you have devoted to the review process.
> >
> > **Q9:** More precise explanation of the smoothness assumption of $p_{X}$ in Equation 2.
> >
> > **A9:** Thanks for raising this point. The smooth function here refers to a function that has continuous derivatives up to second order w.r.t. the Lebesgue measure, i.e., the cross-derivatives needed in Equation 2 always exist. We have added the explanation in the updated manuscript according to your suggestion.
> >
> > **Q10:** More explanation of the solution of the previous work [4] could be introduced in the paragraph following Equation 2 for completeness.
> >
> > **A10:** Thank you very much for this helpful suggestion. We have added a more detailed explanation in that paragraph in light of your suggestion. Basically, this is the general solution of Equation 2 on $\mathbb{R}^n$, where $g, h: \mathbb{R}^{n-1} \rightarrow \mathbb{R}$ are some functions and $n = d$ in our setting.
> >
> > **Q11:** "the gradient is undefined" does not sound to be precise. It should be ‘the gradient doesn’t exist'.
> >
> > **A11:** Thanks so much for your time and helpful advice. We have modified it in the updated manuscript.
> >
> > **Q12:** Why is the limit 5000 nodes in Figure 2? What happens if we use 10,000 nodes?
> >
> > **A12:** We appreciate this question. When we have 10,000 nodes, the running time is 7725 seconds, which has been included in the updated manuscript. Clearly, one can go beyond 5000 variables; we have presented results of running time with variables from 200 to 5000 to illustrate the scalability of our approach, which suggests that the running time is approximately linear w.r.t. the number of variables. At the same time, if we have an extremely large number of variables (e.g., 100,000), besides running time, the computational challenge also lies in the available memory. Several techniques for scalability, such as parallelization and GPU acceleration, could be helpful, which is worth exploring in future applications.

---

> > > ### Author Response · Authors · 2022-11-15
> > > **We appreciate the reviewer for the time dedicated, constructive suggestions, and encouraging feedback (3/4).**
> > >
> > > **Q13:** How strong is the assumption of nondegeneracy?
> > >
> > > **A13:** Thanks so much for this excellent question. The nondegeneracy assumption for score matching is originally introduced in [5] to guarantee the consistency of the estimation, defined in the sense that no different parameters give the same density. In our manuscript, the nondegeneracy assumption is used only to simplify the analysis. If we have multiple parameters that give the same density, we can still get the same characterization matrix for structure recovery. Hence the method still works even in the degenerate setting.
> > >
> > > Besides, although our method does not rely on that, the nondegeneracy assumption itself might not be a strong assumption. It lies in the foundation of score matching on many applications, such as image generation [6], image reconstruction [7], and audio synthesis [8]. Its satisfying performance widely validated in different real-world scenarios suggests that the assumption of nondegeneracy is likely to be held in practice.
> > >
> > > **Q14:** What is the domain of the penalty parameters?
> > >
> > > **A14:** Thanks for the reminder. The domain is [0,1] and we have updated it in the manuscript.
> > >
> > > **Q15:** The usage of “trick” might be informal.
> > >
> > > **A15:** We appreciate your comment and have changed the word to “strategy”. Just for more context, the word ‘trick’ comes from the original paper on score matching [5], which is also the work we referred to in the manuscript. (E.g., see the sentence before Theorem 1 in [5], i.e., “*this is because we can use a simple **trick** of partial integration to compute the objective function very easily*”.) But we agree that it might be informal.
> > >
> > > **Q16:** How to reformulate Equation 8 to Equation 10?
> > >
> > > **A16:** Thanks for raising this point. As mentioned in the manuscript, the basic idea comes from [9], specifically, by replacing the gradient in Equation 8 with a general linear operator $\mathcal{L}$. It is worth noting that $\nabla_{\mathbf{x}} \log p(\mathbf{x}) = \frac{\nabla_{\mathbf{x}} p(\mathbf{x})}{p(\mathbf{x})}$, where $\nabla_{\mathbf{x}}$ denotes the gradient. Of course, one also needs to replace integration with summation and PDF with PMF to get Equation 10. We have added more explanations in the updated manuscript.
> > >
> > > **Q17:** What does the quote imply in saying that $\mathcal{L}$ is “complete”.
> > >
> > > **A17:** We are grateful for this question. Now we have changed the sentence to make it very specific: it is complete in the sense of the definition that follows (specifically, Definition 1 in our manuscript).
> > >
> > > **Q18:** Implications of the proposed framework on nonparametric causal discovery are not very useful.
> > >
> > > **A18:** Thanks for your comment. Accordingly, we have moved this section into the Appendix in the updated manuscript (Appx. C.1). Moving it to the appendix also helps us to get enough space to include proof sketches and other necessary information. Thanks!
> > >
> > > **Q19:** Experimental results about the accuracy of structure learning in Figure 1.
> > >
> > > **A19:** Thanks for the very informative observation. To strengthen the evaluation of our method, we have **re-run all experiments on a set of new datasets**, where the Markov network structures are randomly generated. We can observe from the new results (Figure 2) that our method **outperforms** baselines in most datasets. Compared to the original results in a rather simple setting (Butterfly distributions, Figure 1), we can see a pattern that the advantages of our approach over baselines in terms of structure recovery are more obvious in more complex settings. Meanwhile, the running times of GS, KCI, and SING are significantly longer than that of ours (e.g., 4020.3s for GS and 62.9s for ours, Table 1). Besides, none of these baselines with similar structure recovery performance is able to run when we increase the sample size to 10,000, as shown in Appx. B.2. We believe that the experimental results are sufficient to support our claim since our framework achieves comparable (or even better) recovery results while being much more scalable.
> > >
> > > We thank you once again for the constructive and insightful suggestions. We hope you will find that our response, along with updated manuscripts and new experiments, has properly addressed your comments. Please kindly let us know if there are any further questions or concerns.

---

> > > > ### Author Response · Authors · 2022-11-15
> > > > **We appreciate the reviewer for the time dedicated, constructive suggestions, and encouraging feedback (4/4).**
> > > >
> > > > **References:**
> > > >
> > > > [1] Raskutti, Garvesh, et al. "Model Selection in Gaussian Graphical Models: High-Dimensional Consistency of\boldmath $\ell_1 $-regularized MLE." Advances in Neural Information Processing Systems 21 (2008).
> > > >
> > > > [2] Ravikumar, Pradeep, et al. "High-dimensional covariance estimation by minimizing ℓ1-penalized log-determinant divergence." Electronic Journal of Statistics 5 (2011): 935-980.
> > > >
> > > > [3] Baptista, Ricardo, et al. "Learning non-Gaussian graphical models via Hessian scores and triangular transport." arXiv preprint arXiv:2101.03093 (2021).
> > > >
> > > > [4] Lauritzen, Steffen L. Graphical models. Vol. 17. Clarendon Press, 1996.
> > > >
> > > > [5] Hyvärinen, Aapo, and Peter Dayan. "Estimation of non-normalized statistical models by score matching." Journal of Machine Learning Research 6.4 (2005).
> > > >
> > > > [6] Song, Yang, et al. "Score-based generative modeling through stochastic differential equations." arXiv preprint arXiv:2011.13456 (2020).
> > > >
> > > > [7] Jalal, Ajil, et al. "Robust compressed sensing mri with deep generative priors." Advances in Neural Information Processing Systems 34 (2021): 14938-14954.
> > > >
> > > > [8] Chen, Nanxin, et al. "WaveGrad: Estimating gradients for waveform generation." arXiv preprint arXiv:2009.00713 (2020).
> > > >
> > > > [9] Lyu, Siwei. "Interpretation and generalization of score matching." arXiv preprint arXiv:1205.2629 (2012).

---

> > > > > ### Author Response · Authors · 2022-11-16
> > > > > **Thanks so much for considering our response and updating the recommendation**
> > > > >
> > > > > We are happy that our response and the updated manuscript are helpful. Thanks again for your time and constructive suggestions!

---

### Official Review · Reviewer_nCND · 2022-10-26

**Confidence:** 4
**Correctness:** 4
**Technical Novelty And Significance:** 3
**Empirical Novelty And Significance:** 3
**Recommendation:** 8

**Clarity, Quality, Novelty And Reproducibility:**

The paper is clearly written. Some notation in the paper can be better explained. For example, it was difficult to understand the super script [c], [d] and [m]. After quite a while, I think they are referring to continuous, discrete and mixed. It would be appreciated if author could add a one. On the setup section of the experiment 5, is W_i independent Q_i a typo? Does author mean P_i independent of Q_i ?

On the theory side, author did great work demonstrate various property of the proposed metrics. On the evaluation, I think there are still some room to improve. e.g. evaluating the method with more benchmarks and metrics.

Questions:
1. How does author models the p_theta in the experiment.
2. In section 2.2,  does the model m has to be a normalized distribution? From the equation 4, it does not seem that the partition function Z will be canceled out.
3. on equation 6, is the formula inside log(*) representing p(x_j, z, x_i1; \theta)? What is the motivation to factor it as p(x_j, z | x_i1) * p(x_i1) ?




**Strength And Weaknesses:**

Strength:
1. In the experiment, author demonstrated that the proposed method is efficient and can scale to large number of variables in a network and large number of examples. At the same time, the performance in small domain is comparable to other more expansive baseline.
2. Compared to existing literature, the proposed dependency metrics are generalized to discrete and mixed domain. This generalization not only expand the application of the work, but also leads to a simple structure learning algorithm based on the score matching framework.

Weakness:
1. The experiment is limited. Author has only explored ground truth structure that are r iid pairs of random variables.
* The evaluation would be stronger if author could explore different MN structures, e.g. different tree structures.
*  Currently author is only demonstrating the quality of the learned structure, it would be interesting to know how accurate of the learned Markov Model as well. For discrete Markov Network structure learning literature, e.g. [1], learning algorithm was evaluated based on log likelihood on the test set.  This enables some other application of utilizing the work to perform density estimation on the mixed domain.

[1] Van Haaren, J., & Davis, J. (2021). Markov Network Structure Learning: A Randomized Feature Generation Approach. Proceedings of the AAAI Conference on Artificial Intelligence, 26(1), 1148-1154.



**Summary Of The Paper:**

The paper proposed a method that could learn the structure of Markov Network given the variable domains are mixed. The learning algorithm makes no assumption on the parametric form of the probability distribution. To achieve this, author proposed to learn a density estimator using the score metrics framework. In addition, author proposed a method to estimate the "dependency" among variables with mixed domain from an arbitrary density model. The dependency score allows author 1) to recover the edges in a Markov Network given an arbitrary probability distribution 2) to introduce a differentiable regularization loss term during training to promote graph sparsity.

**Summary Of The Review:**

I like the paper as it demonstrate an elegant solution to learn structure for mixed data. I would like the paper more if the evaluation could be more comprehensive to convince me on its applicability to more complex problem.

==== After rebuttal ===
It is appreciated that author prepared additional experiments to justify the utility of the algorithm in a more complex setting. It does make me feel more confident on the proposed method.

---

> ### Author Response · Authors · 2022-11-15
> **We are very grateful for your time, insightful comments, and encouragement.**
>
> We are very grateful for your time, insightful comments, and encouragement. With gratitude, we have added **a new set of experiments** for all considered evaluations as well as **a new section on detailed experimental settings** and **new results**. The manuscript has also been modified thoroughly in light of all those constructive suggestions. Below please see our point-by-point response below.
>
> **Q1:** Extension of the experiments for a stronger empirical evaluation.
>
> **A1:** We sincerely appreciate this essential point. We have re-run all experiments on a set of new datasets, where the Markov network structures are randomly generated. The settings and results have been included in the updated manuscripts (Section 4 and Appendix B). From these new results (Figure 2), we can observe a similar pattern that the advantages of our approach over baselines in terms of structure recovery are more obvious in complex scenarios (distributions from random graphs) than in rather simple ones (Butterfly distributions). Together with the much better scalability results (Table 1 and Figure 3), we hope you will find that the potential of our method is both theoretically exciting and empirically clear.
>
> We are also grateful for the suggestion on the new evaluation metric regarding the accuracy of the learned Markov model. At the same time, we have not discovered an immediate way to achieve it because it is nontrivial to find the likelihoods even with given score functions (the partition functions are intractable). As you know, one motivation behind score matching-based methods is to avoid intractable partition functions. If we overlooked relevant developments in the literature, please kindly let us know.
>
>
> **Q2:** Notations regarding the data types.
>
> **A2:** Thanks so much for raising this point. We apologize for the confusion and have added a description of them before their first occurrences.
>
> **Q3:** Regarding $W_i$ and $Q_i$ in the setting of Butterfly distribution.
>
> **A3:** Thanks so much for the great catch. $W_i$ should be independent of $P_i$ instead of $Q_i$. We have fixed this typo in the updated manuscript.
>
> **Q4:** How $p_\theta$ is modeled in the experiment.
>
> **A4:** Thanks for raising this point, which helps improve the clarity of the experiment section. We used the deep kernel exponential family (DKEF) in the experiment. Generally, other deep estimation models could also be used depending on various contexts such as computational resources and experimental settings. We have included this detail in the manuscript.
>
> **Q5:** Does the model $m$ in Equation 4 have to be normalized?
>
> **A5:** Thanks for this question. We do not assume that the considered models are normalized. Specifically, let $m’(V, \theta)$ be the normalized PMF, where $\log m(V, \theta) = \log m’ (V, \theta) + \log z(\theta)$ and $V$ denotes the set of variables. In this case, $z$ will be canceled out because we consider only the joint PMF over all variables in Equation 4. We have updated the manuscript to emphasize it.
>
> **Q6:** What is the motivation for factoring the joint density function in Equation 6 as the product of a PDF and a PMF?
>
> **A6:** Thanks for the great question. We adopt this type of factorization in our notation to help represent the joint probability measure in the mixed-type case in a unified and convenient way. In particular, we would like to make a clear distinction between operations on the continuous and discrete data (e.g., how to represent the ‘difference’ in each data type differs).
>
>
> Thank you again for all these insightful comments, which are really helpful to improve the quality of the manuscript. We hope the modified manuscript and new experiments could adequately address the concern. Please let us know if there are any further questions.

---

> > ### Author Response · Authors · 2022-11-16
> > **Thank you so much for your encouragement**
> >
> > We sincerely appreciate the reviewer for checking the response and updated manuscript. Thanks again for your constructive feedback!

---

### Decision · Program_Chairs · 2023-01-20

**Decision:**

Accept: poster

**Justification For Why Not Higher Score:**

While this is a very nice paper I did not see any particular reason for a highlight. But would not be against either.

**Justification For Why Not Lower Score:**

No reason to reject.

**Metareview: Summary, Strengths And Weaknesses:**

All reviewers agree this paper should be accepted. I think the reviews summarize very well the strengths and weaknesses of the paper as well as points that the authors should include in the revised version. I think this will be a great addition to the conference.

**Note From Pc:**

if the above contains the word "oral" or "spotlight" please see: "oral" presentation means -> notable-top-5% and "spotlight" means -> notable-top-25%. As stated in our emails, we are disassociating presentation type from AC recommendations